# Evaluating Compositional Scene Understanding in Multimodal Generative Models

**Shuhao Fu\*, Andrew Jun Lee\*, Anna Wang**
*Department of Psychology,*
*University of California, Los Angeles*

**Ida Momennejad**
*Microsoft Research, NYC*

**Trevor Bihl**
*Air Force Research Laboratory*

**Hongjing Lu**
*hongjing@ucla.edu*
*Department of Psychology, Department of Statistics,*
*University of California, Los Angeles*

**Taylor Webb**
*taylor.w.webb@gmail.com*
*Microsoft Research, NYC*

**\* Equal contribution**

**Reviewed on OpenReview:** `https://openreview.net/forum?id=7bIfe2I7bK`

## Abstract

The visual world is fundamentally compositional. Visual scenes are defined by the composition of objects and their relations. Hence, it is essential for computer vision systems to reflect and exploit this compositionality to achieve robust and generalizable scene understanding. While major strides have been made toward the development of general-purpose, multimodal generative models, including both text-to-image models and multimodal vision-language models, it remains unclear whether these systems are capable of accurately generating and interpreting scenes involving the composition of multiple objects and relations. In this work, we present an evaluation of the compositional visual processing capabilities in the current generation of text-to-image (DALL-E 3) and multimodal vision-language models (GPT-4V, GPT-4o, Claude Sonnet 3.5, QWEN2-VL-72B, and InternVL2.5-38B), and compare the performance of these systems to human participants. The results suggest that these systems display some ability to solve compositional and relational tasks, showing notable improvements over the previous generation of multimodal models, but with performance nevertheless well below the level of human participants, particularly for more complex scenes involving many ($> 5$) objects and multiple relations. These results highlight the need for further progress toward compositional understanding of visual scenes.

## 1 Introduction

A fundamental aspect of human visual processing is its compositional nature. By composing objects and relations in novel ways, we have the capacity to imagine and accurately perceive an infinite variety of visual scenes, even highly improbable ones (Spelke & Kinzler, 2007; Hafri & Firestone, 2021; Cavanagh, 2021).

This ability to exploit the compositional nature of visual scenes is an important priority for the development of general-purpose, human-like computer vision systems.

Recent progress in multimodal generative models has led to the development of systems with an impressively general range of visual capabilities, including systems for text-to-image generation (Radford et al., 2021), and multimodal vision-language models (Achiam et al., 2023). Notably, these systems have been touted for their compositional abilities, as evidenced by their ability to generate novel and unusual scenes depicting, for instance, 'a baby daikon radish in a tutu walking a dog', or the famous 'avocado chair' (Heaven, 2021). On the other hand, these systems also sometimes display basic failures of compositionality. For example, it was reported that the text-to-image model DALL-E 2 was incapable of reliably generating 'a red cube on top of a blue cube' (Ramesh et al., 2022), and followup work found that DALL-E 2 could not reliably generate images to match prompts describing scenes involving relations (Conwell & Ullman, 2022; Marcus et al., 2022). Similarly, while the advent of multimodal vision-language models such as GPT-4 has been heralded as a major step toward the development of systems with general-purpose visual understanding (Wu et al., 2023), some recent studies have identified major shortcomings in their ability to accurately interpret visual scenes, especially for scenes involving the composition of multiple objects and relations (Mitchell et al., 2023; Rahmanzadehgervi et al., 2024).

In this work, we assess the extent to which the current generation of multimodal generative models has made progress toward developing a capacity for compositional image understanding. Our experiments are divided into two broad sections. In the first section, we evaluate the ability of the text-to-image model DALL-E 3 to generate images based on spatial and agentic relations. We first assess the degree of improvement for relational prompts (e.g., 'a blanket covering a box') used in previous work (to test DALL-E 2) (Conwell & Ullman, 2022), and then test novel variants of these prompts, including reversed prompts that evaluate improbable scenarios (e.g., 'a rabbit chasing a tiger'), and compositional prompts involving multiple relations among several entities. We carry out online experiments with external raters to assess the extent to which the generated images accurately depict the objects and relations described in the prompts. In the second section, we evaluate the ability of multimodal vision-language models (GPT-4v, GPT-4o, Claude 3.5 Sonnet, QWEN2-VL-72B, and InternVL2.5-38B) to infer relational patterns from a set of example images. We test problems involving both synthetic images (the Synthetic Visual Reasoning Test (SVRT) (Fleuret et al., 2011)) and real-world scenes (using the Bongard-HOI dataset (Jiang et al., 2022)), and perform a direct comparison with human behavioral data.

Our results indicate that multimodal generative models have indeed made some progress toward compositional scene understanding, with DALL-E 3 displaying improvements over DALL-E 2 in its ability to generate images from relational prompts, and the evaluated multimodal language models all displaying some capacity to infer relational patterns. These successes are tempered, however, by persistent failures of compositionality. While DALL-E 3 is capable of generating images based on more common relational prompts, performance degrades for reversed prompts involving less common scenarios, or prompts involving multiple relations. Similarly, all of the evaluated multimodal language models perform well below human participants in their ability to infer relational patterns, with performance falling to chance levels for problems involving many ($> 5$) objects. Overall, these results suggest that the current generation of multimodal generative models do not possess a robust capacity for compositional scene understanding.

A summary of our contributions is provided below:

- We propose two new datasets for evaluating the robustness of relational text-to-image generation through the use of reversed and compositional relational prompts.

- We perform a comprehensive human behavioral evaluation of relational image generation in a state-of-the-art text-to-image model (DALL-E 3).

- We perform a comprehensive evaluation of relational concept learning in five multimodal language models (GPT-4v, GPT-4o, Claude 3.5 Sonnet, QWEN2-VL-72B, and InternVL2.5-38B), for both real-world and synthetic images, and directly compare the performance of these models with human behavior.

## 2 Evaluating Relational Image Generation in Text-to-Image Models

We evaluated the ability of the text-to-image model DALL-E 3 to generate images based on prompts that described scenes involving relations. We performed three human experiments to evaluate the validity of images generated by text-to-Image models. Section 2.1 describes an experiment evaluating DALL-E 3 on the relational prompts used in Conwell and Ullman's study of DALL-E 2 (Conwell & Ullman, 2022). Section 2.2 reports an experiment involving reversed versions of these prompts, intended to test DALL-E 3's ability to generalize to unusual scenarios (e.g., 'a rabbit chasing a tiger'). Section 2.3 describes an experiment involving prompts with many objects and multiple relations, testing DALL-E's ability to compositionally generalize by combining relations into more complex configurations.

### 2.1 Image Generation with Basic Relational Prompts

#### 2.1.1 Methods

We first assessed the ability of DALL-E 3 to generate images based on basic relational prompts, using the same prompts as those used in a previous study evaluating DALL-E 2 (Conwell & Ullman, 2022). We adopted a design similar to the one used in previous work (Conwell & Ullman, 2022), asking human participants to judge whether images matched the corresponding text prompts.

We used a set of 75 prompts introduced by Conwell & Ullman (2022) to evaluate relational image generation. Each prompt involved two entities with a single relation between them (e.g., 'a tiger chasing a rabbit'). We found that it was necessary to modify some prompts due to policy violations with DALL-E 3 (e.g., prompts involving 'kicking' or 'hitting'). 25 prompts were modified with slight wording changes, while respecting the criteria used to create prompts described by Conwell & Ullman (2022) (see Section A.2). Prompts were broadly classified into those that involved physical relations (in, on, under, covering, near, occluded by, hanging over, and tied to), and those that involved agentic relations (pushing, pulling, touching, chasing, hugging, helping, and hindering). We refer to this set of text prompts adopted from Conwell & Ullman (2022) as *basic relational prompts*.

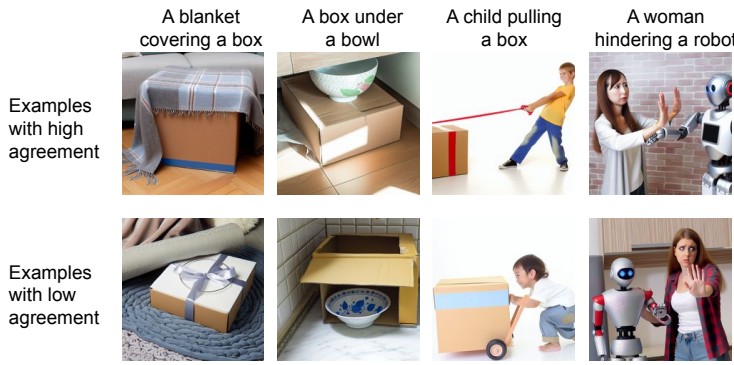

Figure 1: Examples of images generated by DALL-E 3 for basic relational prompts (prompts the describe scenes involving relations), using the 'natural' style. See Figure 14 for more examples.

Images were generated by prompting DALL-E 3 (Betker et al., 2023), a text-to-image model developed by OpenAI, through the Microsoft Azure API[1] (version `2024-02-01` for all experiments). DALL-E 3 was instructed to use the exact prompt provided, and not to revise the prompt before image generation (prompts are sometimes automatically revised by GPT-4 before image generation). The final prompt used by DALL-E 3 was checked against the original prompt, and, if necessary, DALL-E 3 was prompted again, until the original and final prompts were identical. We generated 10 images for each prompt, with the following hyperparameters: quality set to 'standard', style set to 'natural', and image size set to $1024 \times 1024$.

---

[1]https://learn.microsoft.com/en-us/azure/ai-services/openai/

We performed a human behavioral experiment to assess the extent to which the images generated by DALL-E 3 matched the prompts, using a design similar to Conwell & Ullman (2022). Participants were presented with a prompt, along with a collection of 10 images generated by DALL-E 3, and asked to select all of the images that matched the prompt. More details can be found in Section A.3.

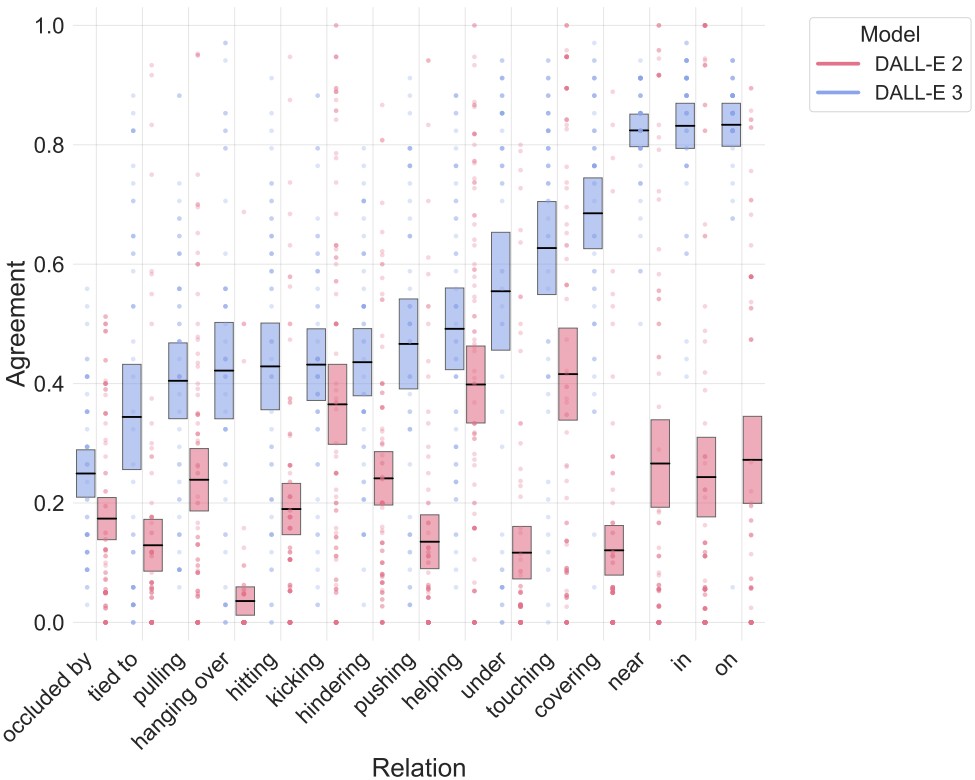

Figure 2: Results for basic relational prompts. The Y axis indicates the agreement between prompts and the images generated by either DALL-E 3 or DALL-E 2, as judged by human participants. Results for DALL-E 3 (in blue) are from the present study. Results for DALL-E 2 (in red) are from Conwell & Ullman (2022). Each point reflects the average agreement for an individual image. Horizontal lines indicate the mean agreement for each relation, and boxes indicate 95% confidence intervals.

### 2.1.2 Results

Figure 2 shows the results for the experiment with basic relational prompts. 'Agreement' reflects the proportion of participants who indicated that the generated images matched with the corresponding prompts, where 1 means that all participants indicated that an image matched the prompt, and 0 means that no participants indicated that an image matched the prompt. Results are grouped according to the specific relations used in each prompt. Results from our experiment with DALL-E 3 are presented in blue, alongside the original results from Conwell & Ullman's (2022) experiment with DALL-E 2 in red.

We found that DALL-E 3 displayed a significantly improved capability, relative to DALL-E 2, to generate images based on basic relational prompts (paired t-test, DALL-E 3 vs. DALL-E 2: $t(49) = 8.552$, $p < .001$), with higher agreement for all relations tested. For some relations in particular, including spatial relations such as 'on', 'near', or 'in', DALL-E 3 displayed major improvements, with average agreement of greater than 0.8, compared with the very low agreement ($< 0.4$) observed in images generated by DALL-E 2. We did not observe any major differences in performance between physical (in, on, under, covering, near, occluded by, hanging over, and tied to) and agentic (pushing, pulling, touching, chasing, hugging, helping, and hindering) relations. Figure 1 shows some examples of the images generated by DALL-E 3.

To assess the extent to which the pattern of results might be influenced by the binary decision in the human behavioral task (i.e., participants only indicated whether or not an image matched the prompt), we also performed an additional experiment involving a more fine-grained rating (a Likert scale from 1 to 7). This experiment yielded qualitatively similar results (see Figure 13 in Section A.3). Overall, the results of our experiments with basic relational prompts indicated improved performance in DALL-E 3 relative to DALL-E 2, with some relations in particular showing major improvements.

## 2.2 Image Generation with Reversed Prompts

### 2.2.1 Methods

Next, we investigated whether DALL-E 3 was able to generate images based on prompts in which the subject and object were reversed. For instance, the basic prompt 'a tiger chasing a rabbit' was changed to 'a rabbit chasing a tiger', switching the roles of the two objects (see Section A.2 for more details about these prompts). The goal of this experiment was to evaluate whether DALL-E 3 was capable of generating relational scenes involving unusual arrangements, as this is typically taken to be a key capacity enabled by compositionality (i.e., unusual scenes can be imagined by combining familiar elements). Images were generated by querying DALL-E 3 in the same manner as the previous experiment, except that the style parameter was set to 'vivid'. We found that DALL-E 3 is more likely to generate matched images for reversed prompts using the 'vivid' style than the 'natural' style. We then performed a behavioral experiment to assess the agreement of the images with their prompts, using a similar design to the experiment with basic relational prompts. Due to the introduction of new policy warnings from reversed prompts, we made 15 more prompt revisions to 75 prompts from Experiment 1, with 7 modifications to already modified prompts and 8 modifications to new ones. This set of 75 was then filtered down to 30 final prompts along with their reversed variants (a total of 60 prompts). More details can be found in Section A.3.

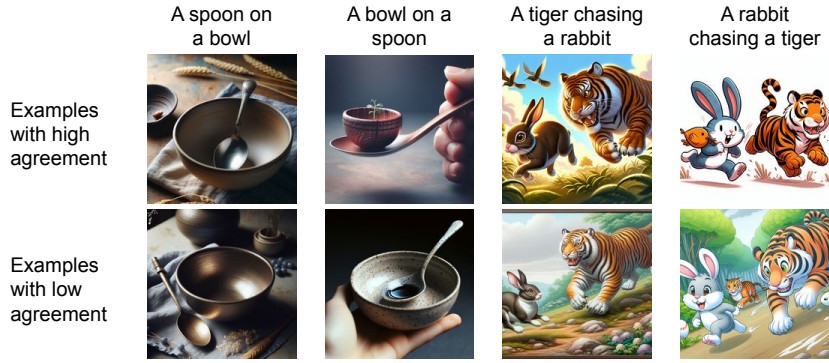

Figure 3: Examples of images generated by DALL-E 3 for basic relational prompts and their corresponding reversed prompts using the 'vivid' style. See Figure 15 for more examples.

### 2.2.2 Results

Figure 2 shows the results for the experiment with reversed prompts. DALL-E 3 displayed significantly worse agreement for reversed prompts than it did for the original basic prompts (paired t-test, basic vs. reversed prompts: $t(29) = 3.668$, $p = 0.001$), though the extent of this effect differed between different relations. Some relations displayed almost no difference in performance between the basic and reversed prompts, including 'near', 'helping', and 'touching', while other relations displayed major difference in performance, including 'on', 'in', and 'chasing'. Notably, two out of the three relations for which performance was not strongly affected by prompt reversal, 'near' and 'touching', are symmetric in nature, so reversing the order of subject and object should not affect the meaning of the prompt, whereas the three relations for which performance was strongly affected, 'on', 'in', and 'chasing', are all asymmetric. It is also worth noting that effect of prompt reversal was particularly pronounced for the relation 'chasing', likely due to the fact that

the reversed prompts described very unusual scenarios (such as a rabbit chasing a tiger). Figure 3 shows examples generated by DALL-E 3 for basic and corresponding reversed prompts.

To systematically assess the impact of prompt likelihood on the validity of the generated images, we measured the plausibility of each text prompt using the GPT-3 language model from OpenAI (the 'davinci-002-1' engine, available through the Microsoft Azure API). We used log likelihood ($LL$), the sum of the log-probabilities of each token in a sentence, as a measure of semantic plausibility (Kauf et al., 2024; Hu & Levy, 2023). We performed a correlation analysis comparing $LL$ with average agreement (between the prompt and the generated images). We found that $LL$ had a statistically significant (but modestly sized) correlation with agreement (spearman correlation $r(598) = 0.090$, $p = 0.027$), consistent with the interpretation that lower agreement for reversed prompts was driven by the lower likelihood of these prompts in DALL-E 3's training data. These results suggest that, although DALL-E 3 shows an improved capability to generate relational scenes relative to DALL-E 2, it is not able to robustly generalize this capability to unusual scenarios that are less likely to be present in the model's training data.

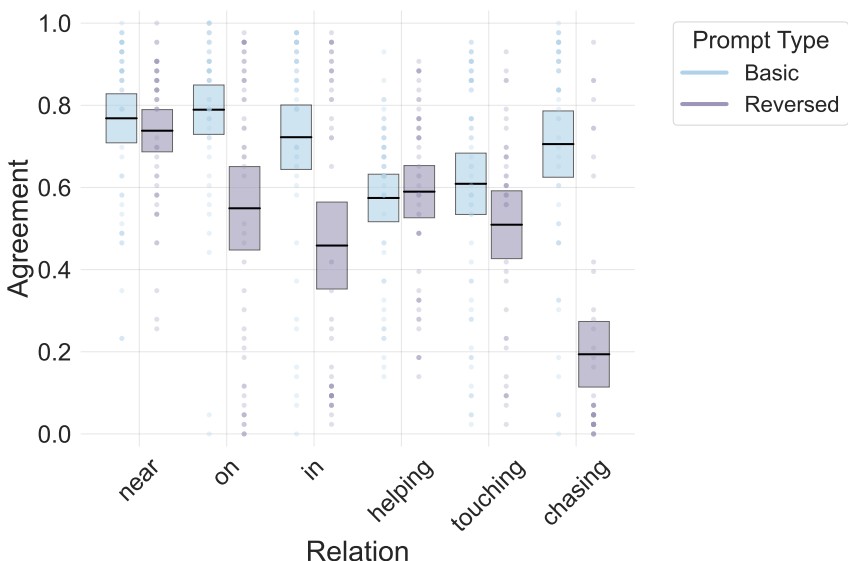

Figure 4: Results for compositional relational prompts. The Y axis indicates the agreement between prompts and the images generated by either DALL-E 3, as judged by human participants. Each point reflects the average agreement for an individual image. Horizontal lines indicate the mean agreement for each relation, and boxes indicate 95% confidence intervals.

## 2.3 Image Generation with Compositional Prompts

### 2.3.1 Methods

Finally, we investigated DALL-E 3's ability to generate images based on more complex prompts involving the composition of two relations. A key feature of image compositionality is the ability to generate new visual scenes by combining multiple objects connected through various relationships. To test this capacity in DALL-E 3, we generated 60 prompts, each of which involved three entities linked by two relations (e.g., 'a spoon tied to a box, with the box near a cylinder'). These prompts were divided into five categories: 10 prompts involved two instances of the same physical relation; 10 prompts involved two different physical relations; 10 prompts involved two instances of the same agentic relation; 10 prompts involved two different agentic relations; and 20 prompts involved one physical relation and one agentic relation. For each prompt, we generated 10 images, yielding 600 images in total. Images were generated by querying DALL-E 3 in the same manner as the previous experiment. We performed a behavioral experiment to assess the agreement of the images with their prompts, using a similar design to the previous experiments (see Section A.3).

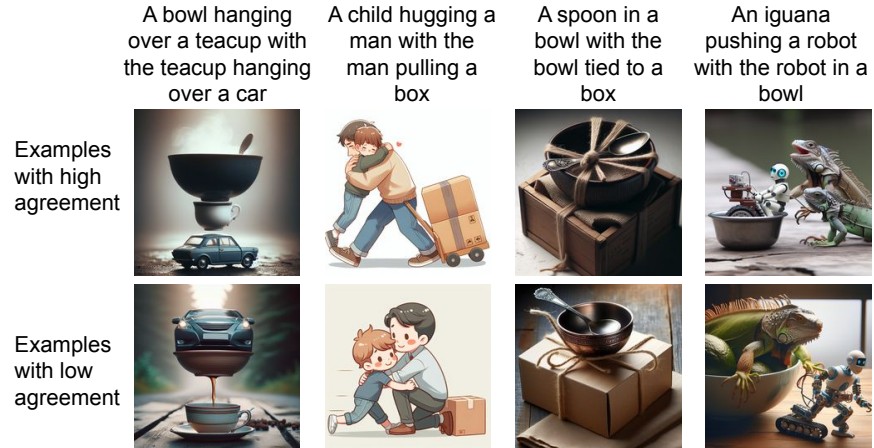

Figure 5: Examples of images generated by DALL-E 3 for compositional relational prompts using the 'vivid' style. See Figure 16 for more examples.

### 2.3.2 Results

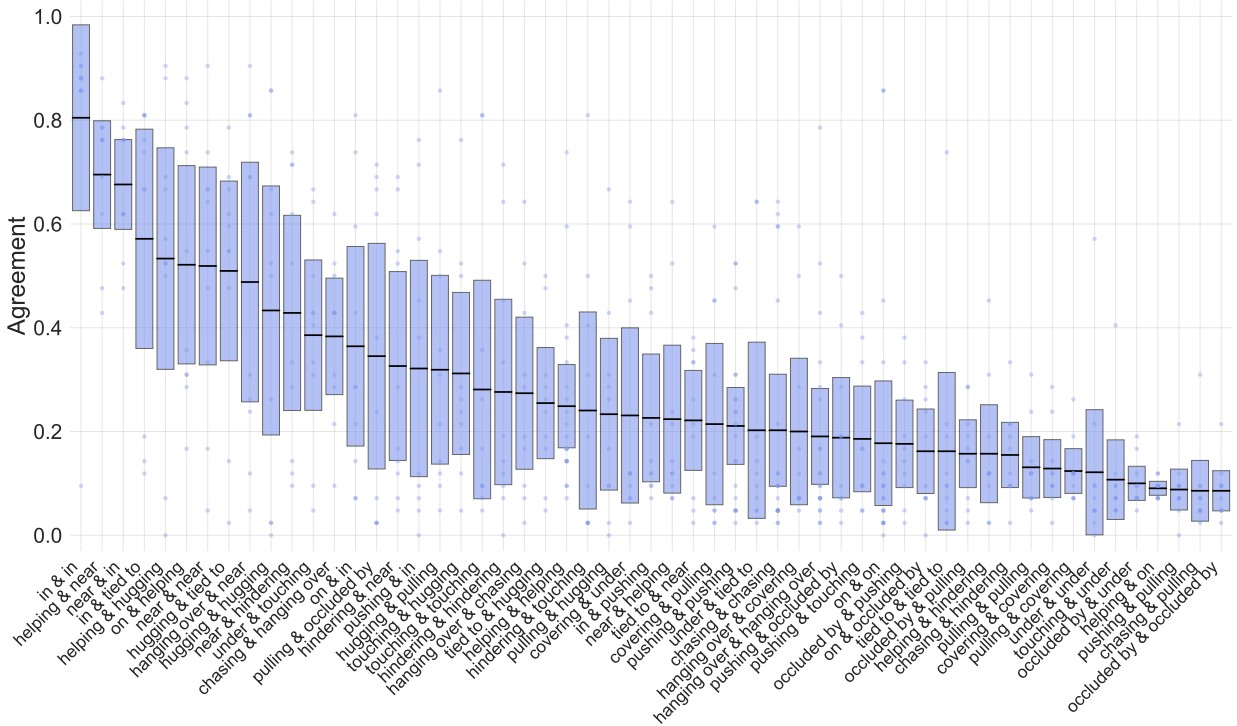

Figure 6: Results for compositional relational prompts. The Y axis indicates the agreement between prompts and the images generated by DALL-E 3, as judged by human participants. Each point reflects the average agreement for an individual image. Horizontal lines indicate the mean agreement for each relation, and boxes indicate 95% confidence intervals.

Figure 6 shows the results for the experiment with compositional relational prompts. Performance varied widely across relation combinations, with some combinations displaying relatively high agreement (e.g. 'in & in' or 'helping & near'). However, the majority of relation combinations displayed relatively low agreement, suggesting that DALL-E 3 had significant difficulty generating images that required the composition of

multiple relations. Figure 5 shows examples generated by DALL-E 3 with text prompts involving multiple objects and relations.

## 2.4 Summary of Relational Image Generation Results

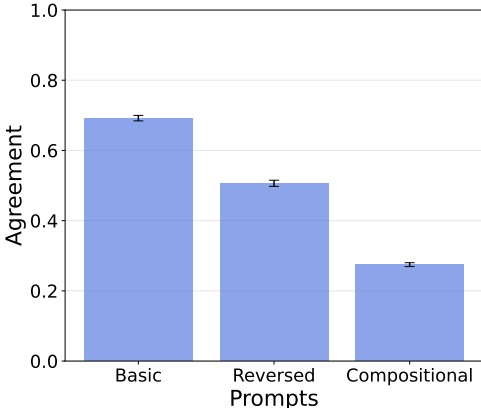

Figure 7: Summary of results for relational image generation experiments with DALL-E 3. Error bars reflect 95% confidence intervals.

Figure 7 shows a summary of the results for the relational image generation experiments. These results highlight two major findings. First, DALL-E 3 displays a notably improved ability to generate relational scenes relative to DALL-E 2. However, DALL-E 3's performance is degraded significantly for prompts in which the order of subject and object are reversed (often yielding unlikely scenes), or prompts involving the composition of two relations. These failure modes indicate that DALL-E 3's capacity for scene generation is not robustly compositional, as a compositional approach would enable entities to be combined in novel ways, and more complex scenes to be formed through the composition of multiple relations.

## 3 Evaluating Relational Concept Learning in Multimodal Language Models

We evaluated the ability of two multimodal variants of GPT-4, gpt-4-vision-preview and gpt-4o, to perform few-shot relational concept learning. We evaluated these models using two datasets: Bongard-HOI (Jiang et al., 2022), involving action- and event-based relations (human-object interaction) in naturalistic scenes, and the Synthetic Visual Reasoning Test (SVRT) (Fleuret et al., 2011), involving visuospatial relations in synthetically generated images. Both datasets test the ability to learn abstract relational concepts from a relatively small number of demonstrations. To ensure that the learned concepts are genuinely relational, these datasets employ 'hard negatives' – incorrect answers that share object-level features with the relational concept, but differ at the relational level (e.g., for the concept 'human rides bicycles', a hard negative will involve a human and a bicycle with a different relation).

### 3.1 Learning Relational Concepts from Real-World Scenes

#### 3.1.1 Methods

We first evaluated relational concept learning with real-world scenes, using the Bongard-HOI dataset (Jiang et al., 2022). This dataset contains 167 distinct concepts involving human-object interactions. Each concept is defined by a subject-verb-object triplet, where the subject is always a human, the verb indicates an action (e.g., 'hold' or 'inspect'), and the object is a common item (e.g., 'bicycle' or 'apple'). Each concept is illustrated by 7 positive examples and 7 negative examples, each presented in a separate image (Figure 8). In the standard evaluation methodology, 6 labeled images from each class are presented (12 total), and the remaining positive and negative images are presented for classification.

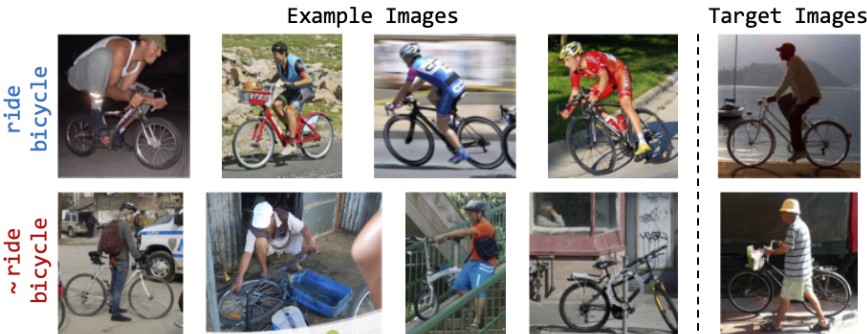

Figure 8: Example relational concept from Bongard-HOI. Positive instances feature a human riding a bicycle, while negative instances feature a human and a bicycle with a different relation.

We evaluated gpt-4-vision-preview and gpt-4o on problems taken from the Bongard-HOI test set, and compared their performance to human performance as reported with the original dataset, as well as the HOITrans baseline model (the best performing model in that work) (Jiang et al., 2022). These models were evaluated using the Microsoft Azure API. Due to the limited number of images that could be presented in the context window for these models (10 images maximum), evaluation was performed by presenting 9 labeled example images (randomly selecting either 5 positive examples and 4 negative examples, or vice versa) followed by a single query image (randomly selecting either a positive or negative example). Given the large size of the dataset (13,941 problems), we sampled 10 problems per concept for our evaluation (1,670 problems in total). Some problems resulted in content policy violations. After excluding problems that triggered policy violations, gpt-4-vision-preview was tested on 1,149 problems (involving 161 concepts) and gpt-4o was tested on 1,156 problems (involving 160 concepts). Temperature was set to 0 when evaluating both models, top-p was set to 1, and the 'detail' parameter was set to 'high'.

### 3.1.2 Results

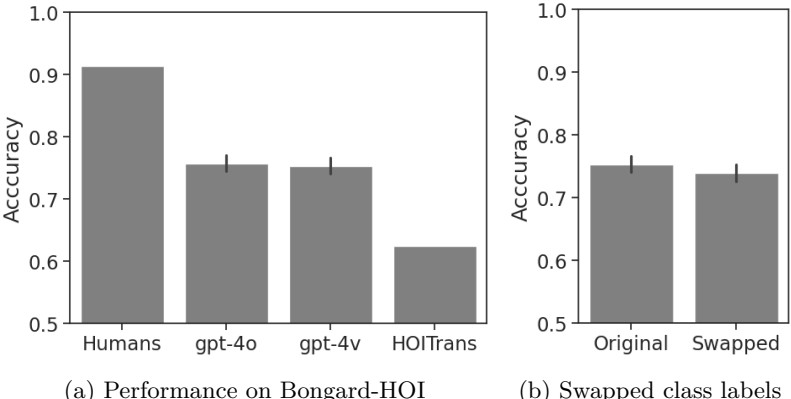

(a) Performance on Bongard-HOI    (b) Swapped class labels

Figure 9: Bongard-HOI results. (a) Few-shot classification accuracy for two variants of GPT-4 (gpt-4o and gpt-4-vision-preview, shortened to gpt-4v) vs. best previous model (HOITrans) and human performance. (b) Performance of gpt-4v on original problems vs. problems with swapped class labels. Error bars reflect binomial 95% confidence intervals.

Figure 9a shows the results for our experiments with the Bongard-HOI dataset. Both GPT-4 variants significantly outperformed HOITrans, the best performing model from previous work (one sample proportion tests, gpt-4o vs. HOITrans: $\chi^2(1) = 80.06, p < 0.0001$, gpt-4-vision-preview vs. HOITrans: $\chi^2(1) = 75.27, p < 0.0001$), but performed significantly worse than the human participants on this task (one sample

proportion task, human vs. gpt-4o: $\chi^2(1) = 335.78, p < 0.0001$, human vs. gpt-4-vision-preview: $\chi^2(1) = 352.61, p < 0.0001$).

One possible concern with these results is that the Bongard-HOI dataset is publicly available on the internet, and therefore may in principle have been present in the training data for these models. To determine whether performance on this task depends on prior training, we performed an experiment in which the labels assigned to classes were swapped (positive examples, with a label of 1, became negative examples, with a label of 0, and vice versa). We tested gpt-4-vision-preview on the swapped-labels task, and compared performance to the standard version of the task (Figure 9b). There was no difference in performance for these two versions of the task (logistic regression, swapped labels vs. original labels: $z = 0.83, p = 0.41$), indicating that performance on this task was not due to memorization of a publicly available dataset. However, it is also important to consider that this dataset features common relational configurations (e.g., a man riding a bicycle) that are likely to have been present in some form in the training data for these models. Thus, while we have ruled out the possibility of literal memorization, there is still a concern that performance may depend to some extent on general familiarity with the relations and configurations present in these problems. To address this, in the next section, we present results from a synthetic dataset (SVRT) with novel relational concepts not likely to be present in the training data.

## 3.2 Learning Relational Concepts from Synthetic Scenes

### 3.2.1 Methods

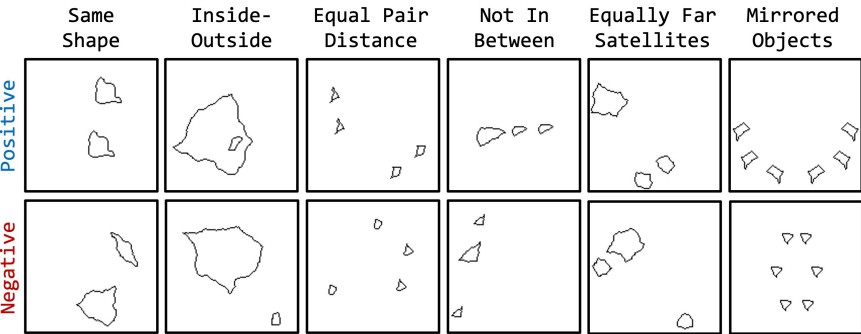

Figure 10: Examples of relational concepts in SVRT. Each concept shows a positive and negative instance.

We evaluated relational concept learning with synthetic scenes using the SVRT dataset (Fleuret et al., 2011). This dataset features abstract relational concepts instantiated with synthetic figures, making it difficult to rely on previous experience with common real-world relational concepts. Additionally, many SVRT problems feature relational concepts involving a larger number of objects and relations, making it possible to test for the ability to learn more complex relational concepts.

The SVRT dataset consists of 23 distinct relational concepts, each involving between 2 and 6 objects. These concepts range from simple pairwise relations (e.g., same shape) to more complex configurations of multiple objects and relations (e.g., mirrored objects) (Figure 10). We used the original codebase from Fleuret et al. (2011)[2] to generate novel positive and negative instances for each concept. This allowed us to ensure that the images would not have been present in the models' training data.

We evaluated gpt-4-vision-preview and gpt-4o on these problems using the Microsoft Azure API. To evaluate a broader set of models, we also evaluated Claude Sonnet 3.5 (through the Anthropic API), and two open-source VLMs: QWEN2-VL-72B and InternVL2.5-38B (these were evaluated locally on a workstation with 4 NVIDIA GPUs). For each problem, we presented 1-9 few-shot examples, consisting of a mixture of positive and negative instances (in random order). Each example was presented in a separate image, and was accompanied by a message indicating the class (0 or 1). After presenting the few-shot examples, we presented a target image (randomly sampling either a positive or a negative instance) and prompted the

---

[2]https://fleuret.org/cgi-bin/gitweb/gitweb.cgi?p=svrt.git;a=summary

model to make a classification. For each relational concept (out of 23), and each number of few-shot examples (1-9), we evaluated each model 10 times, using completely different images each time, resulting in a total of 2,070 tests. The models occasionally made formatting errors in its responses. In these cases, we re-prompted the model until it produced a correctly formatted response. Temperature was set to 0 when evaluating both models, top-p was set to 1, and the 'detail' parameter was set to 'high'.

We also performed a comparison with human behavioral data from Lee et al. (2023). In that experiment, participants were presented with SVRT problems in interactive episodes. For each episode, participants received a series of positive and negative examples illustrating a relational concept, and attempted to classify each example. After each response, they received feedback indicating the correct classification. The previous examples from the current episode remained on screen, and were sorted based on class. The episode continued until participants made 7 correct classifications in a row. See Lee et al. (2023) for more details on the human behavioral experiment, and Section A.7 for more details on the analysis comparing VLMs with human performance.

### 3.2.2 Results

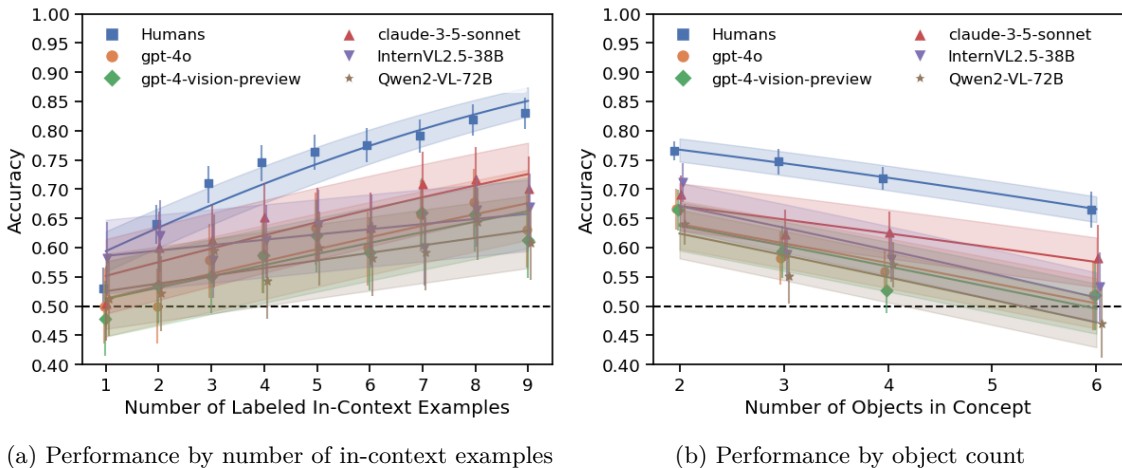

(a) Performance by number of in-context examples      (b) Performance by object count

Figure 11: SVRT few-shot classification accuracy for humans, gpt-4o, gpt-4-vision-preview, claude 3.5 sonnet, Qwen2-VL-72B, and InternVL2.5-38B by (a) number of in-context examples and (b) number of objects in the concept. Points are averages across all SVRT concepts and attempts, coupled with binomial 95% confidence intervals. Curves are logistic regression fits with binomial confidence interval bands. Dashed black lines indicate chance performance.

Figure 11 shows the comparison of human participants with GPT-4 on the SVRT. Human participants significantly outperformed all models (logistic regression, gpt-4-vision-preview: $\beta = 0.63, z = 12.01, p < 0.0001$; gpt-4o: $\beta = 0.66, z = 12.65, p < 0.0001$; claude 3.5 sonnet: $\beta = 0.44, z = 8.18, p < 0.0001$; Qwen2-VL-72B: $\beta = -0.70, z = 13.52, p < 0.0001$; InternVL2.5-38B: $\beta = -0.52, z = 9.82, p < 0.0001$). Both human participants and all VLMs displayed improved performance with more in-context examples (human: $\beta = 0.17, z = 15.57, p < 0.0001$, gpt-4-vision-preview: $\beta = 0.078, z = 4.47, p < 0.0001$, gpt-4o: $\beta = 0.085, z = 4.88, p < 0.0001$; claude 3.5 sonnet: $\beta = 0.096, z = 5.32, p < 0.0001$; Qwen2-VL-72B: $\beta = 0.053, z = 3.08, p = 0.0021$; InternVL2.5-38B: $\beta = 0.038, z = 2.18, p < 0.030$). (Figure 11a), and degraded performance as a function of the number of objects in a relational concept (human: $\beta = -0.13, z = 6.18, p < 0.0001$, gpt-4-vision-preview: $\beta = -0.18, z = 5.12, p < 0.0001$, gpt-4o: $\beta = -0.16, z = 4.69, p < 0.0001$; claude 3.5 sonnet: $\beta = -0.1144, z = 3.29, p = 0.0010$; Qwen2-VL-72B: $\beta = -0.16, z = 4.61, p < 0.0001$; InternVL2.5-38B: $\beta = -0.19, z = 5.62, p < 0.0001$) (Figure 11b). Notably, performance for both GPT-4 variants and the two open-source VLMs was statistically indistinguishable from chance levels for problems with 6 objects (gpt-4-vision-preview: $\chi^2(1) = 0.3, p = 0.58$; gpt-4o: $\chi^2(1) = 0.3, p = 0.58$; Qwen2-VL-72B: $\chi^2(1) = 0.83, p = 0.36$; InternVL2.5-38B: $\chi^2(1) = 1.070, p = 0.30$), although Claude Sonnet showed performance above chance for these problems ($\chi^2(1) = 6.85, p = 0.0089$). Thus, while the VLMs that

we investigated displayed some capacity to learn relational concepts in this task, and was sensitive to the same factors as human participants (number of in-context examples and number of objects per image), performance was nevertheless well below human level, and they were generally not capable of solving more complex problems involving a greater number of objects and relations. In the Appendix, we also present additional experiments that explore the role of chain-of-thought prompting (section A.8), and interactive testing of GPT-4 (section A.8.1), finding that neither of these factors improve performance on this task.

## 4 Related Work

### 4.1 Datasets for Evaluating Compositionality in Neural Networks

A number of previous studies have investigated the ability of neural networks to process compositional scenes. Prior to the development of large-scale pre-trained models, several datasets were proposed for evaluating compositional visual processing and reasoning in deep learning models, including Visual Genome (Krishna et al., 2017), CLEVR (Johnson et al., 2017), PGM (Barrett et al., 2018), and OOD (Geirhos et al., 2021), typically emphasizing out-of-distribution and compositional generalization (but with large, task-specific training sets).

### 4.2 Evaluating Compositionality in CLIP and Text-to-Image Models

More recent research has focused on evaluating the zero-shot and few-shot capability of pre-trained multi-modal generative models to process and generate compositional scenes. Lewis et al. (2022) investigated the compositional processing capabilities of Contrastive Language-Image Pre-Training (CLIP) (Radford et al., 2021), on which both text-to-image models and multimodal language models are based, finding that CLIP could compose concepts at the single-object level, but could not reliably compose multiple objects or relations. Marcus et al. (2022) and Conwell & Ullman (2022) investigated the ability of DALL-E 2 to generate images based on compositional and relational prompts. Our study adds to this work by investigating the ability of DALL-E 3 to generate compositional scenes, finding that it has improved performance relative to DALL-E 2, but is still not robustly compositional, especially in settings with many objects and relations. Concurrent work from Conwell et al. (2024) also shows that DALL-E 3 struggles with prompts that involve logical operators such as negation.

### 4.3 Visual Reasoning in Multimodal Language Models

Other recent studies have investigated the ability of multimodal language models (e.g., GPT-4v) to process compositional scenes. Both Yiu et al. (2024) and Mitchell et al. (2023) evaluated GPT-4v on visual analogy problems. Yiu et al. (2024) found limited success (GPT-4v was able to solve visual analogies for certain types of relations), whereas Mitchell et al. (2023) found that GPT-4v performed very poorly on problems from the Abstraction and Reasoning Corpus (ARC) (Chollet, 2019), even for problems that human participants perform nearly perfectly on. Our work adds to these studies by investigating the ability of these models to infer relational concepts in a few-shot setting (whereas analogy involves the inference of a relational pattern from just a single example), finding limited success in this domain.

It is also worth noting that previous work has found that both convolutional neural networks and vision transformers can perform nearly perfectly on the SVRT with a very large number of training examples (Kim et al., 2018; Messina et al., 2021b;a). In this work we were interested specifically in the ability to solve SVRT problems with a relatively small number of examples (as is done with human participants), which remains a challenging setting (Vaishnav & Serre, 2022; Webb et al., 2024b; Mondal et al., 2024).

## 5 Discussion

In this work, we have presented a broad-ranging evaluation of compositional scene understanding in multi-modal generative models. In our experiments with text-to-image models, we found that DALL-E 3 shows an improved capability to generate relational images relative to the previous generation of these models (DALL-

E 2), but displays a lack of robustness for images involving reversed (i.e., novel) relational configurations, and more complex scenes involving many objects and more than one relation. In our experiments with multimodal generative models, we found that all of the models we evaluated (including two multimodal variants of GPT-4, Claude 3.5 Sonnet, and two open-source models) displayed some capability for few-shot learning of relational visual concepts, for both real-world scenes and abstract visuospatial problems, but underperformed relative to human participants, and did not show any capacity to learn relational concepts involving a larger number of objects. Overall, these results present a complex picture, with the current generation of multimodal generative models displaying some capacity for compositional and relational processing, while still falling short of the robustness and generality exhibited by human visual processing.

These results raise the question of how to reconcile the complex pattern of successes and failures observed both in the present set of experiments and other recent work. A major theme throughout all of these studies is that the current generation of models seem to display especially poor performance in more complex multi-object settings. Lewis et al. (2022) found that CLIP was capable of composing concepts at the single-object level, but could not do so reliably for multi-object scenes. This is echoed by our findings with text-to-image models, where performance is especially degraded for prompts involving a large number of objects. In the case of multimodal generative models, both our results and the results of Yiu et al. (2024) indicate some capacity for processing of visual relations, whereas Mitchell et al. (2023) find very poor performance on the ARC dataset. This discrepancy may also be explainable by a difficulty with multi-object processing: the SVRT and Bongard-HOI (both investigated in this work), and KiVA (investigated by Yiu et al. (2024)) all involve a relatively small number of objects per image, whereas ARC problems typically involve a much larger number of objects. In the few cases that we investigated that did involve a larger number of objects (for some SVRT problems), performance was very poor, with only one out of the five models that we evaluated showing performance better than chance.

Interestingly, similar results have also been observed for tasks that do not involve relations, such as counting (Rane et al., 2024; Rahmanzadehgervi et al., 2024), or multi-object scene description (Campbell et al., 2024). Indeed, Campbell et al. (2024) found that multimodal language models were able to solve visual analogy problems when they were decomposed into object-level representations, but could not do so reliably when the same problems were presented within a single image. This pattern of results suggests that the poor performance observed in the current generation of multimodal models may be due primarily to a more basic difficulty with parsing of multi-object scenes, rather than a difficulty with relational processing per se. This would also explain the discrepancy between the relatively poor performance of multimodal models on visual analogy problems and the more robust capability of language models on text-based analogy problems (Webb et al. (2023), Musker et al. (2024), Webb et al. (2024a), cf. Lewis & Mitchell (2024)).

Finally, it is worth considering how the compositional and multi-object capabilities of multimodal generative models might be further improved. Campbell et al. (2024) argued that these difficulties are related to the classic 'binding problem' from cognitive science (Treisman & Gelade, 1980; Greff et al., 2020; Frankland et al., 2021), which arises any time that a shared set of representational resources must be used to represent multiple objects, but without a mechanism for binding features at the level of objects. In recent years, object-centric representation learning methods, most notably including slot-based approaches (Burgess et al., 2019; Locatello et al., 2020), have arisen as a solution to the binding problem, and have been shown to dramatically improve performance in visual reasoning tasks that require multi-object compositionality (Ding et al., 2021; Mondal et al., 2023; Webb et al., 2024b; Mondal et al., 2024). This suggests that a promising direction for future work may be to combine multimodal generative models with object-centric visual processing methods.

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

# A  Appendix

## A.1  Code and Data Availability

All experimental materials, code, and data are available at:

> https://github.com/andrewjlee0/evaluating_compositionality_VLMs

## A.2  Prompts for Relational Image Generation Experiments

To test relational image generation in text-to-image models, we used a set of 75 relational prompts developed by Conwell & Ullman (2022). Due to policy violations identified by DALL-E 3, we made the following modifications to these prompts:

- 'kicking' → 'hugging'

- 'hitting' → 'chasing'

- 'knife' → 'spoon'

- 'a teacup near a cylinder' → 'a teacup near a box'

- 'a monkey touching an iguana' → 'a monkey touching a teacup'

- 'a cylinder near a bowl' → 'a teacup near a bowl'

- 'a man hindering a man' → 'a man hindering a robot'

- 'a woman pulling an iguana' → 'a monkey pulling an iguana'

- 'a child pulling a box' → 'a robot pulling a car'

- 'a monkey hindering a monkey' → 'a monkey hindering an iguana'

- 'a man helping a man' → 'a man helping a monkey'

- 'a robot helping a robot' → 'a robot helping a man'

This list describes all modifications for both Experiments 1 and 2. In Experiment 1, we sought to re-test the original list of 75 prompts but changed 25 due to policy violations. In Experiment 2, we modified 15 of these 75 (7 of the already 25 modified ones in Experiment 1 and 8 new ones, resulting in 33 total prompts with at least one modification) to deal with reversed prompts that yielded policy violations, or prompts for which the reversed version was identical (e.g., 'a man helping a man'). From this set of 75, we then selected 30 prompts and their corresponding reversed versions for Experiment 2. Specifically, we selected the three physical and agentic relations for which DALL-E 3 displayed the highest level of agreement when generating images based on basic prompts. The selected physical relations were 'near', 'on', and 'in'. The selected agentic relations were 'touching', 'helping', and 'kicking'. Because prompts with the relation 'kicking' frequently resulted in policy violations, we replaced 'kicking' with 'chasing'. For each of these 6 relations, we used 5 basic prompts, and 5 reversed prompts in which the subject and object were swapped, resulting in a total of 60 prompts (30 basic and 30 reversed). For each prompt, we used DALL-E 3 to generate 10 images, yielding 600 images in total.

### A.3 Details for Human Behavioral Evaluation of Generated Images

We performed three human behavioral experiments to assess the extent to which the images generated by DALL-E 3 matched the prompts. To evaluate images generated from basic relational prompts (Section 2.1), we recruited 34 participants (26 female, average age = 20.18 years). Four of these participants were excluded from analysis because they indicated a lack of seriousness in the post-experiment survey. To evaluate images generated from reversed relational prompts (Section 2.2), we recruited 43 participants (38 female, average age = 19.6 years). One participant was excluded for taking too long to complete the experiment (> 3000 minutes), and four of these participants were excluded because they indicated a lack of seriousness in the post-experiment survey. To evaluate images generated from compositional relational prompts (Section 2.3), we recruited 42 participants (34 female, average age = 20.9 years). Five participants were excluded because they indicated a lack of seriousness in the post-experiment survey. All participants were recruited through the [redacted for anonymity] Psychology subject pool, and were compensated with course credit. All behavioral experiments were approved by the [redacted for anonymity] Institutional Review Board.

All experiments were conducted remotely via an online platform (Figure 12). On each trial, participants were presented with a prompt, along with a collection of 10 images generated by DALL-E 3, and asked to select all of the images that matched the prompt (Figure 12). Although there was no time limit, participants were encouraged to respond as quickly and accurately as possible. There were 75 trials for experiment 1 (basic relational prompts), 60 trials for experiment 2 (reversed prompts), and 60 trials for experiment 3 (compositional prompts). Trials were presented in a random order for each participant.

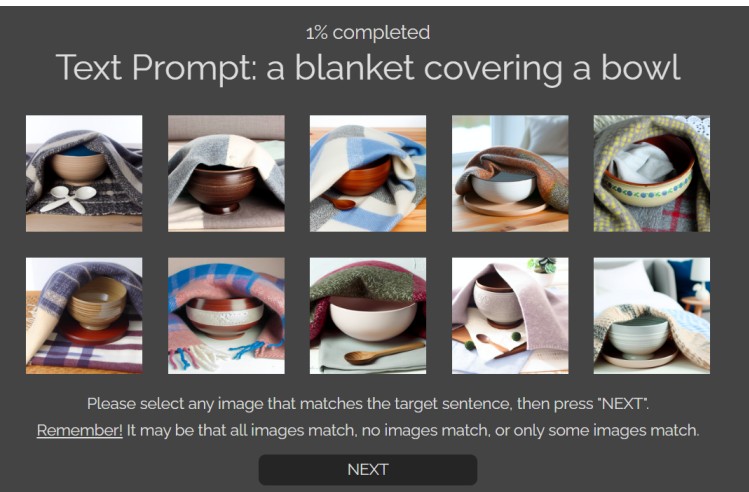

Figure 12: An example of the display presented to participants in the behavioral experiment.

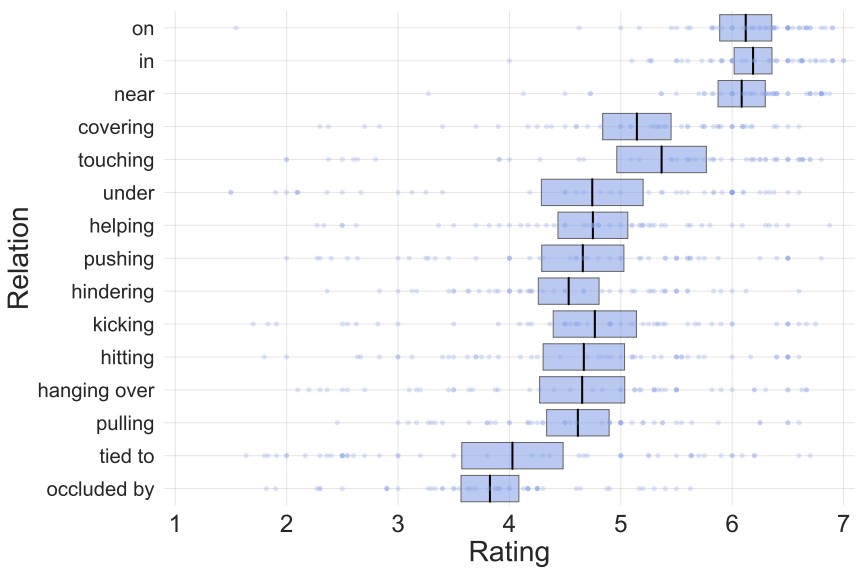

Figure 13: Results for basic relational prompts when participants responded using a Likert scale from 1 to 7 instead of making a binary judgment. Relations are sorted along the Y axis based on the rank order of agreement in the binary decision paradigm (in Figure 2). These results illustrate that the overall qualitative pattern (i.e., the rank ordering of the relations) is very similar between the two paradigms. Each point reflects the average rating for an individual image. Vertical lines indicate the mean rating for each relation, and boxes indicate 95% confidence intervals.

We also conducted an additional behavioral experiment in which participants assessed images using a Likert scale (from 1 to 7) rather than making a binary judgment. We recruited 45 participants for this study (37 female, average age = 19.84 years). Three participants were excluded from the analysis because they indicated a lack of seriousness in the post-experiment survey. Each participant completed a total of 750 trials, each of which included a single image generated from a basic relational prompt. The results of this experiment were qualitatively similar to the experiment that used a binary decision (Figure 13), so we used the binary decision paradigm for the experiments with reversed and compositional prompts.

### A.4 Additional Examples of Images Generated by DALL-E 3

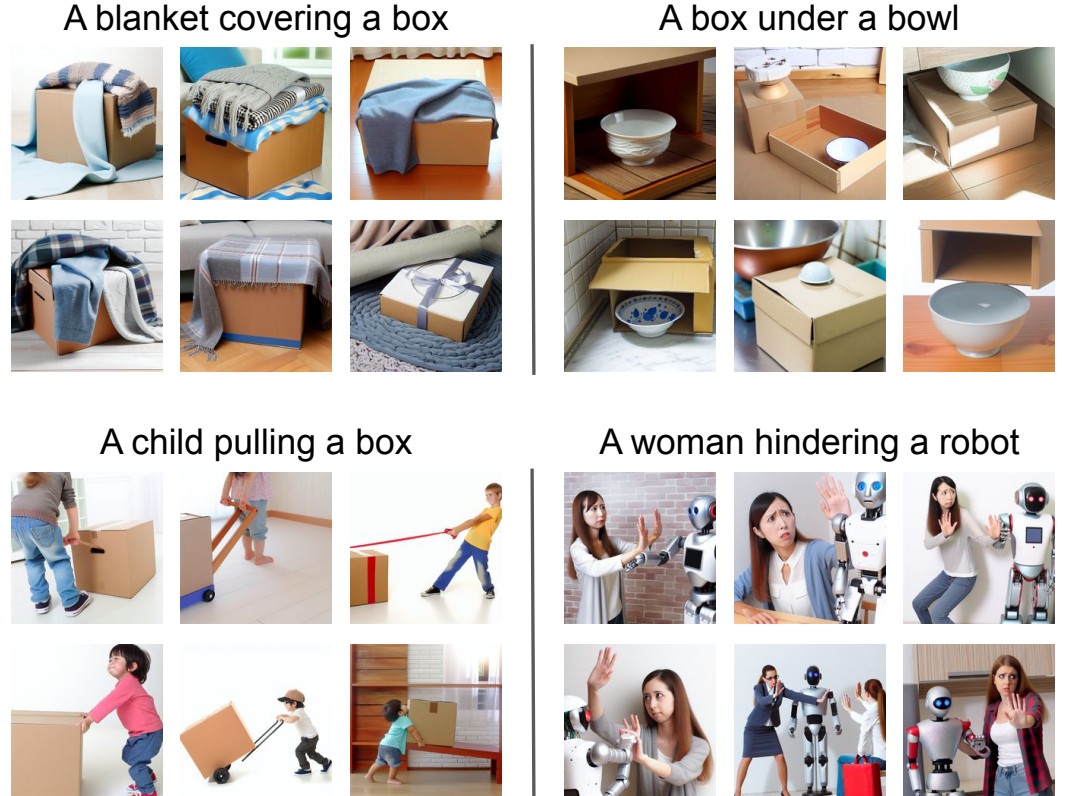

Figure 14: Additional examples of images generated by DALL-E 3 for basic relational prompts using the 'natural' style.

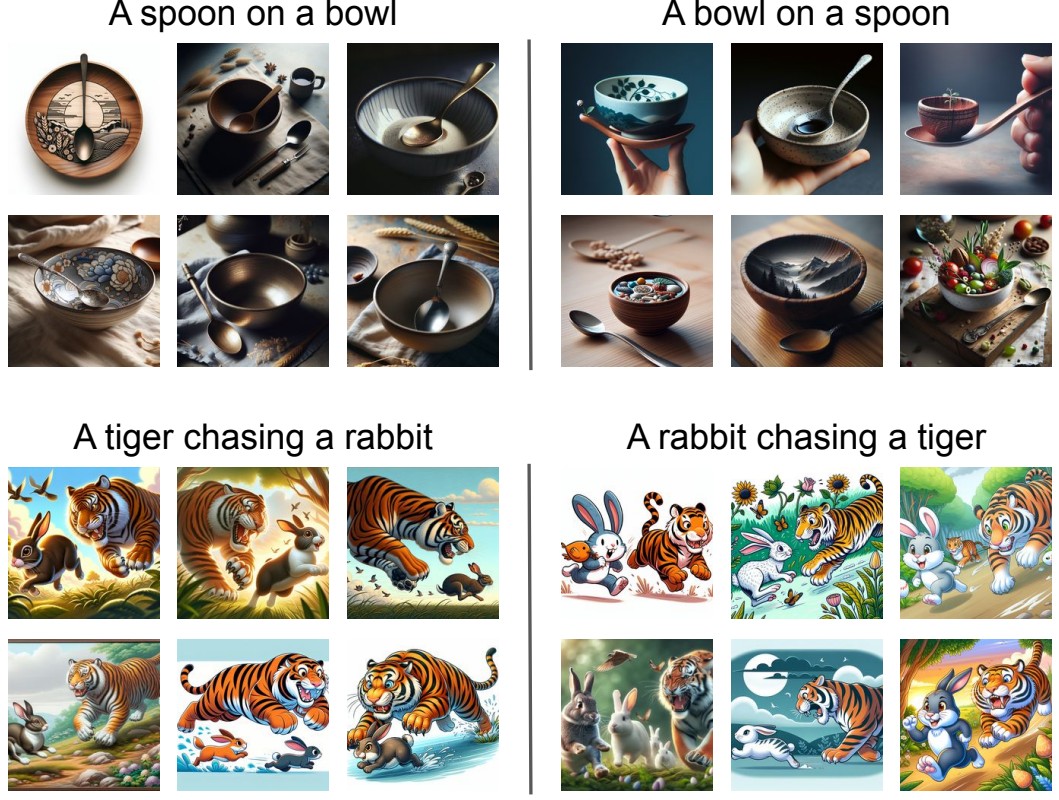

Figure 15: Additional examples of images generated by DALL-E 3 for basic relational prompts and their corresponding reversed prompts using the 'vivid' style.

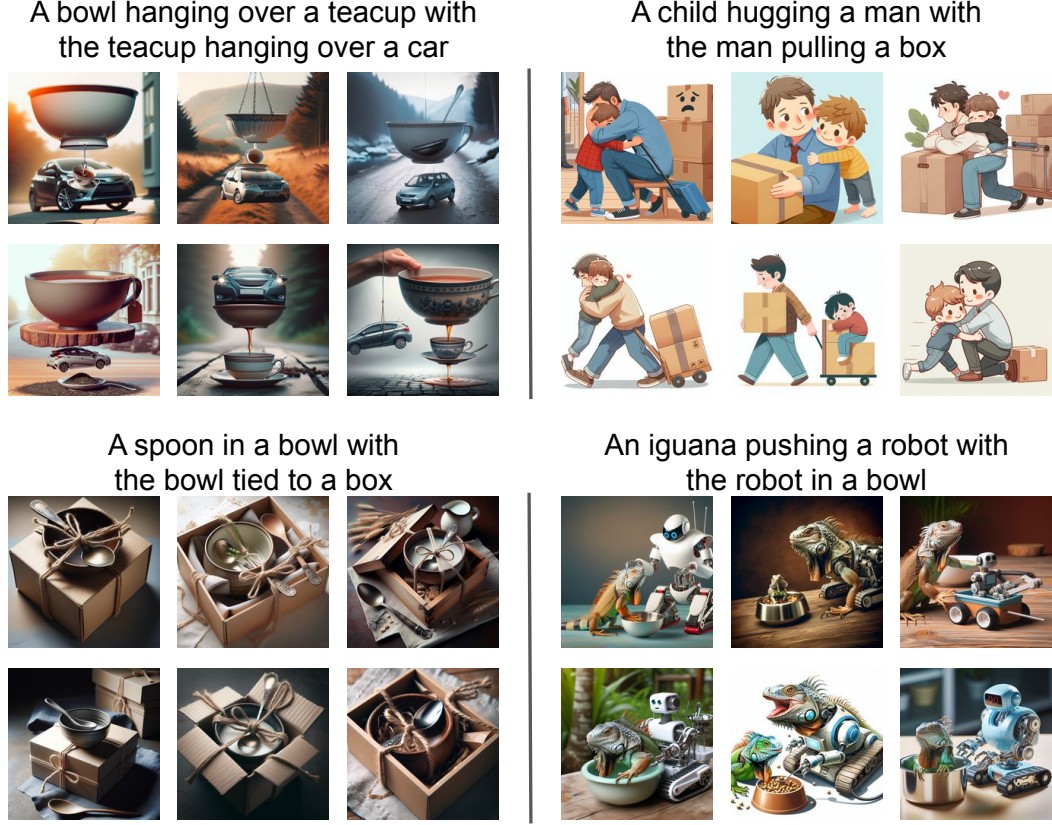

Figure 16: Additional examples of images generated by DALL-E 3 for compositional relational prompts using the 'vivid' style.

### A.5 Analysis of Errors in Images Generated by DALL-E 3

We performed an analysis to better understand the types of errors found in the images generated by DALL-E 3. This analysis focused on the images generated for reversed relational prompts. We inspected each of the images with an average agreement of less than 0.2, and categorized them according to the following types of errors: wrong subject, wrong object, wrong relation, and relation binding error. For example, given the prompt 'a rabbit chasing a tiger', a wrong subject error would indicate that the image does not contain a rabbit, a wrong subject error would indicate that the image does not contain a tiger, a wrong relation error would indicate that the image does not contain a 'chasing' relation, and a relation binding error would indicate that the roles are reversed, i.e. the image depicts a tiger chasing a rabbit. The results (Figure 17) showed that the overwhelming majority of errors involved either a wrong relation or a relation binding error.

### A.6 Using Multimodal Language Models to Evaluate Images Generated by DALL-E 3

We performed an analysis to determine whether multimodal language models could be used to automatically evaluate the images generated by DALL-E 3, focusing on the generated images for basic relational prompts. To do so, we prompted each model to rate whether a generated image matched the original prompt on a scale from 0 to 10. We then performed a correlation analysis between these scores and the average score for each image from the human behavioral experiment. As a baseline, we estimated the inter-subject reliability of the human ratings. This was accomplished by randomly splitting the human participants into two groups and computing the correlation between these groups. We performed this simulation 100 times and computed the average of the correlation scores obtained in each run.

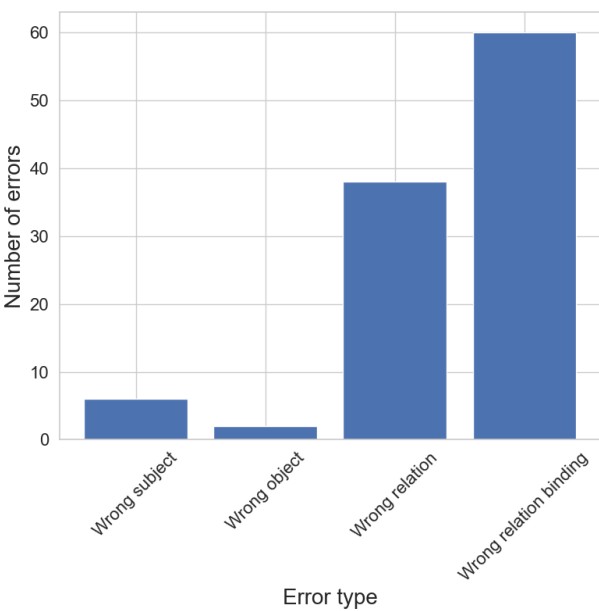

Figure 17: Error analysis for images generated by DALL-E 3.

This analysis revealed that the human participant ratings were highly correlated with each other (human-to-human correlation: $r = 0.91$), whereas the models showed much lower correlations with the human ratings (GPT-4v: $r = 0.49$; GPT-4o: $r = 0.51$; Claude 3.5 Sonnet: $r = 0.43$; Qwen2-VL-72B: $r = 0.56$; InternVL2.5-38B: $r = 0.48$). These results indicate that multimodal language models cannot reliably be employed to automate the assessment of images generated by text-to-image models. This is consistent with the results of our evaluation of multimodal language models, which showed that they suffer from limited understanding of compositional images comparable to the limited ability of text-to-image models to generate those images.

### A.7 Comparison of Multimodal Language Models with Human Behavioral Data on SVRT

We compared a set of multimodal language models (including GPT-4o, GPT-4-vision-preview, Claude 3.5 Sonnet, QWEN2-VL-72B, and InternVL2.5-38B) with human performance on SVRT for each number of few-shot examples. In the behavioral experiment from Lee et al. (2023), episodes were terminated after participants made 7 correct classifications in a row. As a result, in some episodes, participants received fewer than 9 examples before the episode was terminated. For the purposes of comparison, we assume that participants in these episodes would have made correct classifications for subsequent examples. For instance, if a participant made 7 correct classifications for the first 7 examples (thus terminating the episode), we assume that they also would have made a correct classification for an 8th and 9th example. This assumption was necessary to avoid biasing the distribution for 8 and 9 examples toward lower performing participants, since the best performing participants were more likely to terminate episodes early by classifying all examples correctly. In Section A.8.1, we present an experiment in which GPT-4 was evaluated using the same paradigm as human participants, allowing for a closer comparison.

### A.8 Evaluating GPT-4 with Chain-of-Thought Prompting

In the main text, we evaluated GPT-4 on the SVRT and Bongard-HOI by presenting a set of labeled in-context examples followed by a target image for classification. Here, we also present results for experiments that used more extensive 'chain-of-thought' prompting strategies (Wei et al., 2022), in which language model output is used to perform intermediate reasoning steps. We investigated four separate Chain-of-Thought conditions, involving different amounts of intermediate computation:

- **Chain-of-Thought 0:** No Chain-of-Thought is performed (this corresponds to the default results presented in the main text).

- **Chain-of-Thought 1:** In-context examples were presented one at a time, each coupled with a revised message instructing GPT-4 to describe the contents of the image. These descriptions were then appended to the context window. The final target/query message was the same as the Chain-of-Thought 0 condition.

- **Chain-of-Thought 2:** Descriptions were not solicited for the in-context examples, but the target/query prompt was expanded to encourage compositional thinking across all images. Specifically, GPT-4 was instructed to think carefully about the objects and relations in the series of labeled examples, about the pattern of relations that may determine category membership, and to apply the patterns to the target.

- **Chain-of-Thought 3:** The combination of prompts for Chain-of-Thought 1 and 2.

The results of these experiments are shown below. On Bongard-HOI, the Chain-of-Thought 0 condition performed as well or better than any of the other conditions (Figure 18). On SVRT, there were no discernible differences between the different Chain-of-Thought conditions (Figure 19). Taken together, these results suggest that performance on these tasks cannot be improved via Chain-of-Thought prompting. This is consistent with the interpretation that performance in these models is constrained by a basic difficulty with multi-object perception, rather than downstream reasoning processes.

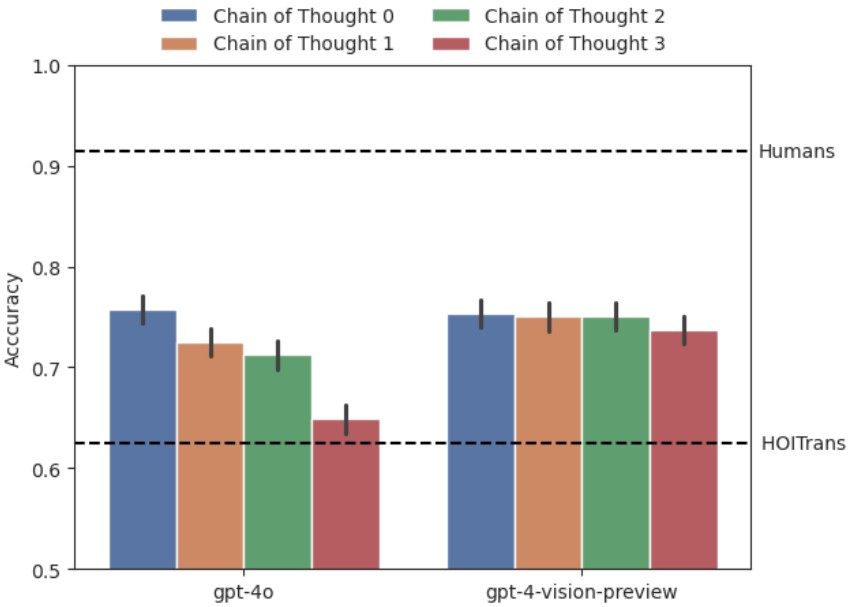

Figure 18: Bongard-HOI Chain-of-Thought results. Dashed lines represent accuracies for humans and the HOITrans baseline (Jiang et al., 2022). Error bars reflect binomial standard errors.

### A.8.1 Interactive Testing of GPT-4 on SVRT

We also performed an alternative evaluation of GPT-4 on the SVRT involving an active learning paradigm, similar to the paradigm that is typically used with human participants for this task. In that paradigm, GPT-4 was prompted to categorize each in-context example one at a time, receiving immediate feedback after each classification. The model maintained a context window of the previous 9 classifications, which allowed it to learn from past errors. On each trial, GPT-4 was presented with an image and asked to categorize it as either positive or negative. After each classification, the model received feedback ('Correct!' or 'Incorrect!') and

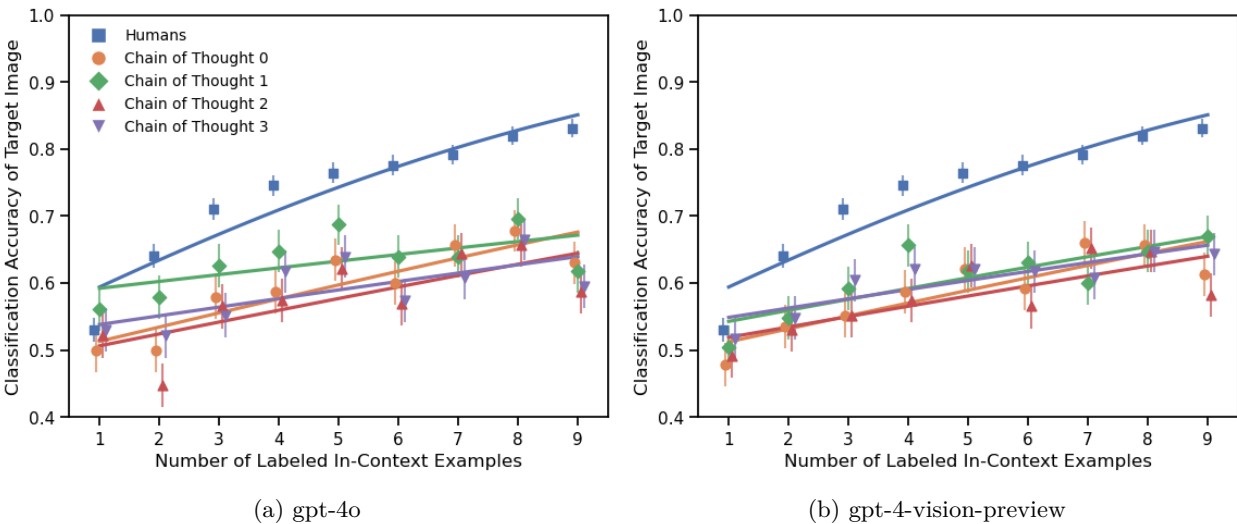

(a) gpt-4o

(b) gpt-4-vision-preview

Figure 19: SVRT Chain-of-Thought results for humans, (a) gpt-4o, and (b) gpt-4-vision-preview. Points are averages across all SVRT concepts and attempts per concept, coupled with binomial standard error bars. Curves are separate logistic regression fits to each chain-of-thought condition.

acknowledged the feedback before moving on to the next image. As with human participants, this process was continued until there were 7 correct classifications in a row or the maximum of 34 attempts was reached.

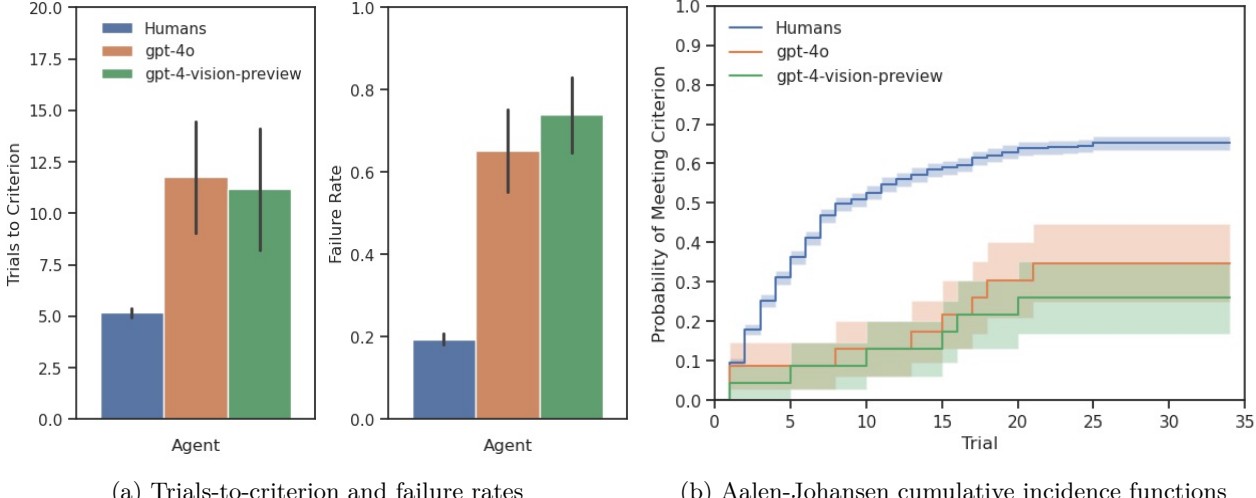

(a) Trials-to-criterion and failure rates

(b) Aalen-Johansen cumulative incidence functions

Figure 20: Active learning and survival analysis. (a) Trials-to-criterion across all non-failed SVRT concepts and failure rates across all concepts with standard error of the mean and binomial standard error, respectively. (b) Probabilities of meeting criterion by trial, or cumulative incidence functions, estimated by Aalen-Johansen survival analysis, with binomial standard error bands.

Two dependent variables were measured per SVRT concept: concept failure (whether criterion was met within the maximum 34 attempts) and trials-to-criterion (no data if concept failure; or total number of classification attempts before 7 correct in a row if criterion met). Figure 20a displays average trials-to-criterion over non-failed concepts and failure rates over all concepts.

We fit a linear and logistic mixed-effects models to failure and trials-to-criterion data respectively, with participant type (human vs. gpt-4o vs. gpt-4-vision-preview) as a fixed effect and problem ID as a random effect. Pairwise contrasts with Tukey-adjusted p-values of 3 estimates reveal that humans failed less fre-

quently (gpt-4o: $z = 5.14, p < 0.0001$; gpt-4-vision-preview: $z = 5.67, p < 0.0001$) and met criterion faster than than both variants (gpt-4o: $t(625) = 3.96, p = 0.0003$; gpt-4-vision-preview: $t(624) = 3.31, p = 0.0028$).

Because trials-to-criterion data are missing for concepts where criterion was not met (i.e., concept failure), average trials-to-criterion underestimate true rates of learning. However, precise estimates of learning speeds can be established using survival analysis, a method that takes into account censored data (e.g., the frequency of concept failures due to a ceiling of 34 attempts) to estimate trial-by-trial probabilities of meeting criterion. A standard approach to survival analysis is to use a Kaplan-Meier estimator for such trial-by-trial probabilities. However, Kaplan-Meier assumes that censorship (i.e., concept failure) is independent of the probability of meeting criterion. Since our experiment violates this assumption by design, we used an Aalen-Johansen estimator with trials-to-criterion as the event of interest and concept failure (censorship) as a *competing risk.*

Figure 20b displays Aalen-Johansen probability curves. At least half of human participants meet criterion by the 9th trial, whereas gpt-4o and gpt-4-vision-preview have a 13% and 8.7% probability of meeting criterion by the same time. At trial 25, 6.5 people out of 10 meet criterion while neither variant reaches 50% of meeting criterion. To compare probability curves, we fit a Fine-Gray competing risks regression with *agent* as a fixed effect and *SVRT concept ID* as a stratified predictor. The Fine-Gray regression with humans as the reference level reveals that humans reach criterion faster than either GPT-4 variant (gpt-4o: $SHR = 0.25, 95\%CI : 0.13 - 0.47, p < 0.0001$; gpt-4-vision-preview: $SHR = 0.18, 95\%CI : 0.087 - 0.38, p < 0.0001$).

## A.9  Testing with Swapped Labels on SVRT

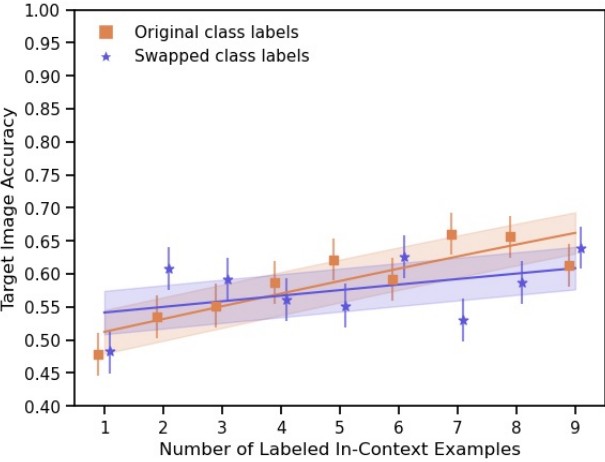

Figure 21: SVRT few-shot classification accuracy for gpt-4-vision-preview with and without swapped class labels.

Finally, as with Bongard-HOI, we tested whether GPT-4's performance can be explained by memorization of the SVRT dataset. Although all SVRT images were generated from scratch for our experiments, we confirmed that memorization could not explain our results by re-testing gpt-4-vision-preview with swapped class labels. There was no significant difference between performance with original vs. swapped labels (Figure 21; logistic regression: $z = 0.87, p = 0.38$).

