# OpenReview forum: "Evaluating Compositional Scene Understanding in Multimodal Generative Models"
_TMLR — Accepted by TMLR_

### Review · Reviewer_Lrk1 · 2024-12-16

**Summary Of Contributions:**

- The work provides a thorough empirical study of complex relational scene understanding and composition for multimodal models, and compares this to a qualitative study using human participants.
- The study evaluates the relational scene understanding of generative image models (DALL-E 2, DALL-E 3) with relational prompts, which are then reversed to generate novel scene configurations the model is less likely to have seen examples of. The study reveals improvement between model versions, but also reveals examples where ...
- The study also evaluates GPT-4 variants on few-shot learning of relational scene understanding and composition on real world (Bongard-HOI) and synthetic scenes (SVRT).
- The work reveals several cases where models score significantly lower than human participants for complex multi-object scenes.
- From the results, the authors provide interesting observations and discussion around the implications of the study, suggesting that some of the observed limitations may stem from difficulties in parsing objects in complex multi-object scenes.
- To alleviate the observed shortcomings, the authors propose integration of object-centric processing methods in representation learning setups, relating the issue to the "binding problem" in cognitive science.

**Audience:**

Yes

**Broader Impact Concerns:**

As already mentioned, the lack of transparency in closed-source models represents an issue which readers may not necessarily be aware of, given the broadening of readership beyond academic circles. However, by adding additional experiments on open-source models, this can be discussed naturally in the text without requiring a broader impact statement.

In a broader context, the study is more an effort towards potentially mitigating relational and compositional biases in scene understanding. It is, in this reviewers opinion, therefore not necessary to extend the paper with a larger broader impact statement, as the paper highlights implicit limitations by the nature of the study itself.

**Claims And Evidence:**

Yes

**Requested Changes:**

While the study is thorough and shows diligence, the issues raised by this reviewer is mostly focused on the question of generality; namely *are the observed results particularities in closed-sourced OpenAI models*, or *are these general limitations of multimodal AI models*?

A set of requested changes are outlined below. We emphasise that **we do not expect a similar scope on any additional experiments as the main results**.

- (C1) With regards to W1, given the non-zero likelihood of bootstrapped training between these models, it would be prudent to investigate the extent of any potential overlap between the evaluated models. In other words, **does poor performance in generating scenes with specific prompts correlate with weak annotations of such scene examples**? This could potentially also serve to show shared cross-modal limitations between the model representations. The paper’s discussion strongly emphasises that poor multi-object parsing might underpin the observed failures across tasks, including text-to-image generation and relational annotation. This aligns with our hypothesis; **there could be deeper dependencies across these modalities due to similar training biases**. Again, I want to point out that such a study does not necessarily need to be similar in scope as the original experimental results. A more limited test could help prospective readers in interpreting the results, and serves to illuminate potential blind spots in reading the paper.
- (C2) Given the scope of the work, it seems unreasonable to ask for an equally broad comparison with open-source models. However, a study of more limited scope using open source alternatives seems advisable, if not completely necessary, to ensure the significance of the study is preserved. This reviewer would therefore recommend **a smaller scale study with open source alternatives** to ensure that the observed phenomena are general limitations of scene understanding in multimodal models. While this reviewer does not believe the results to be invalid, or non-transferrable to open-source models, showing that results extend beyond models trained by the same group / lab / company serves to emphasise the significance to the reader, and would strengthen the work.

**Strengths And Weaknesses:**

**Strengths**:

- (S1) The work represents an interesting study of the capabilities of state-of-the-art models on complex scene understanding, incorporating multimodal perspectives through both text-to-image generation, and image-to-text annotation.
- (S2) The work is thorough, extensive, and well grounded in qualitative studies with human participants, emphasising potential gaps in model performance and human performance on complex visual scene understanding tasks. These types of study are important for understanding current limitations in state-of-the-art multimodal models in common use today.
- (S3) Following S2, the results show gaps in model performance compared to human abilities in scene understanding. In particular, the gap between performance in Figures 6, 9, 11, were (in this reviewers opinion) quite striking. As such, the study serves as an informative read for modellers, and can serve to better illuminate potential gaps in multimodal learning signals. The authors highlight this in their discussion, and propose (at a very high level) more object-centric inductive biases or priors for representation learning paradigms.
- (S4) The authors show thoroughness in the discussion of some of the experiments by pointing out that since it is not presently known what data was used to train the models, the results for Bongard-HOI could be a result of data leakage, i.e. the models could have, in fact, been trained on the dataset it is being evaluated on, which in turn, limits the significance of the results. Expanding the study to SVRT, and noting that this task results in a drop in performance, shows thoroughness and care in interpreting results and designing experiments.
- (S5) The authors seem committed to reproducibility by stating their intent to release the code.
- (S6) The appendix includes several additional results. The lack of improvement by chain-of-thought prompting, as well as the section on interactive testing both served as interesting reading.

**Weaknesses**
- (W1) While the study is comprehensive, the study focuses exclusively on closed-source models from OpenAI. While this is understandable given the ubiquitous use of DALL-E and GPT-4, one should consider that all models are trained by the same lab / company, and biases that may stem from standardised in-house modelling approaches may result in similar results and limitations observed in both text-to-image and image-to-text models. It is not unlikely (read; highly likely) that these models have been trained in conjunction with each other using a bootstrapping approach. Given that image-to-text and text-to-image models are essentially producing inputs for the other, this seems to be a natural distillation / tuning step. However we can only speculate as to what extent such a bootstrapping method was applied in the final product. Granted, the authors do not make expressive comparisons between the models from different modes, but this will likely affect the diversity of the results, and may lead to potential blind spots in the study.
- (W2) Following the same line of argument, it is difficult to know exactly how significant parts of the study can reasonably be expected to be, particularly in the case for the image-to-text models. As the authors point out, it is difficult to use publicly available datasets for evaluation, as it is likely the data could have been sourced for training the model. For S4, I commend the authors in highlighting this issue. However, the generality of the study is somewhat inhibited by the choice to focus on closed-sourced OpenAI models. It also limits the practical applicability for the general community, since it is not clear that the results can be interpreted as general issues in multimodal modelling, or as a result of unknown confounding effects that could stem from particularities with OpenAI's training pipelines. While this could seemingly be considered tangential, broadening the study by showing similar characteristics (naturally, on a smaller scale) would alleviate any doubts that these are general phenomena, not inherent in strictly closed-source AI products available to the public as a product.

---

> ### Author Response · Authors · 2024-12-22
>
> Dear Reviewer Lrk1,
>
> Thank you very much for the thoughtful feedback and suggestions. We are currently in the process of performing additional experiments to address the issues raised, and will provide a detailed response soon.
>
> Sincerely,
> Authors

---

> ### Author Response · Authors · 2025-01-17
>
> We would like to thank the reviewer once again for the incredibly thoughtful review and helpful suggestions. We have now implemented several followup experiments to address the concerns raised, and have revised the manuscript accordingly (with revisions tracked in blue). Below we provide a point-by-point description of the changes:
>
> ## Evaluation of proprietary models from other labs
>
> We completely agree that it is important for the evaluation to include models that are not all from the same lab. To address this, we have now run Claude Sonnet 3.5 on the SVRT dataset (for which we could be most certain that neither the dataset nor similar variants were present in the training data). The results were qualitatively similar to those obtained with GPT-4v and GPT-4o. Specifically, we found that Claude Sonnet was affected by similar factors as human participants – i.e. performance improved as a function of the number of examples provided, and performance decreased as the number of objects in a rule increased – but the overall performance was nevertheless significantly below human participants. We did find that Claude Sonnet performed somewhat better than GPT-4o (with an overall accuracy increase of 4-5%), and performed better than chance on problems with 6 objects, but the overall pattern was very similar to that observed for GPT-4v and GPT-4o, suggesting that all of these models suffer from similar limitations which aren’t likely to result from idiosyncratic differences in training data between labs.
>
> We have included these results in Figure 11 and section 3.2.2 of the revised manuscript.
>
> ## Evaluation of open-sourced models
>
> We also agree that it is important to include open-source models in our evaluation. To address this, we have now evaluated both QWEN2-VL-72B and InternVL2.5-38B on the SVRT dataset. We found that both of these models showed similar performance profiles to the GPT-4 models – i.e. they were both similarly sensitive to the number of examples and the number of objects – and both of these models performed significantly worse than human participants. The two open-source models also performed at a similar level as GPT-4o (i.e., there were no statistically significant differences between these models).
>
> These results are also now included in Figure 11 and section 3.2.2.
>
> ## Evaluation of generated images with multimodal language models
>
> The reviewer raised the concern that common failure modes observed between DALL-E 3 and GPT-4v/o might result from training on the same data distribution. An alternative explanation is that these commonalities arise from fundamental challenges that are faced by all multimodal generative models (whether text-to-image or image-to-text) when processing scenes involving multiple objects and images (i.e., the `binding problem’ that we consider in the discussion). To adjudicate between these two explanations, we prompted multimodal language models to rate the images generated by DALL-E 3, and performed a correlation analysis comparing the responses to human judgments. As a baseline, we randomly divided the human participants into two groups and performed a correlation analysis comparing their responses to each other. We find that the ratings generated by all models, including GPT-4v/o, the open source models (QWEN2 and InternVL) and Claude Sonnet, are significantly less correlated with human ratings than the human-to-human baseline. Furthermore, the GPT-4 models were neither most correlated nor least correlated with human ratings. QWEN2 showed the highest correlation (though still much lower than the human-to-human baseline), and Claude Sonnet showed the lowest correlation. That is, all of the multimodal language models that we tested are relatively bad at rating the agreement between captions and images generated by DALL-E 3, to a roughly equal extent, and this does not appear to be related to whether the models were developed by OpenAI. These results are most consistent with fundamental limitations facing all of these models (given similar overall architecture and training procedures, particularly a lack of object-centric processing as considered in the discussion) rather than idiosyncratic details of the training data distribution employed by one lab.
>
> These results are now included in the Appendix, section A.6.
>
> We would like to thank the reviewer once again for the very helpful comments and suggestions. We believe that the paper is stronger thanks to the inclusion of these additional tests, and look forward to any additional feedback or questions.

---

### Review · Reviewer_CS2x · 2024-12-24

**Summary Of Contributions:**

*contributions*
1. While improving compositional image generation is not new, the paper offers a systematic evaluation and pinpoints where models still struggle.
2. Better compositional reasoning is a big deal for practical uses—everything from design workflows to storytelling can benefit. The analysis tackles real-world scenarios in generating images with multiple objects and relationships.

*presentation*
1. The paper clearly explains why compositional tasks matter and how the images are graded. The visual examples make it much easier to spot where things go right (or wrong).
2. Each section flows nicely, clearly laying out how DALL·E 3 differs from earlier generative models.
3. The side-by-side image comparisons and prompt examples are exceptionally well done, making it straightforward to judge whether the compositional instructions have been followed.

*evaluation*
1. The paper employs a combination of several clear-cut evaluation methods such as human judgments, the Likert scale, and binary compositional accuracy. Probably the most salient results include:
1. The use of human judges ensures that the final scores align well with real-world understanding and preferences.
2. Real-world data, from Bongard-HOI, but also synthetic data like SVRT featuring flipped and compositional
prompts offer holistic views of model relational reasoning.
3. Finally, the authors experiment with log-likelihood as a proxy to investigate how well the model's internal probability estimations reflect its actual performance, thus shedding light on possible biases in the training data.

*main strength*
1. This paper systematically tests a wide range of prompts covering spatial relations, object properties, and contextual interactions.
2. The authors detail where the model stumbles—such as mixing up relationships when things get complicated, losing track of numbers, or drifting toward more common images in its training data.
3. By pitting DALL·E 3 against DALL·E 2 and involving human evaluators, the study paints a clear picture of both the progress made and where improvements are still needed. The visual examples really drive these points home

**Audience:**

Yes

**Claims And Evidence:**

Yes

**Requested Changes:**

Address the weakness and question to authors

More hybrid metrics should be developed that would measure both accuracy and creativity.

Provide fine-grained analyses of failure cases, especially involving spatial or relational mismatches.

**Strengths And Weaknesses:**

*Main Concerns*
1. Most of the statements tested were literal and descriptive; fewer abstract or metaphorical requests, such as "happiness climbing a staircase of dreams," might see whether the model is able to deal with deeper or more symbolic themes.

2. The paper doesn’t focus much to scenarios involving Counterfactual or other logical constructs, such as “an elephant reading a book while not standing on a surfboard.” Testing such prompts could offer more insight into DALL·E 3’s capacity for abstract reasoning.

3.  Less obvious mistakes in very complex prompts are still difficult to find. There is high reliance on human evaluations; this raises scalability issues. Large-scale testing requires more automation.

*Questions for Authors*
1. In which way does the evaluation process take place? Are there any standard
labeling on which judges verify whether every element is aligned according to the text, or does it rely on a summary subjective opinion?
2. How does this hold up against unusual prompts or highly specialized fields, like medical imaging or intricate diagrams?
3. With the much more restrictive content filters of DALL·E
3, does this restrict its capabilities in handling compositional instructions for borderline or sensitive prompts?
4. Test prompts like "an elephant reading a book while not standing on a surfboard" require an understanding of negation. How would one go about effectively evaluating such cases, especially when training
data does not extensively see similar scenarios?

---

> ### Author Response · Authors · 2024-12-30
>
> Dear Reviewer CS2x,
>
> Thank you very much for the thoughtful feedback and suggestions. We are currently in the process of performing additional experiments to address the issues raised, and will provide a detailed response soon.
>
> Sincerely, Authors

---

> ### Author Response · Authors · 2025-01-17
>
> We would like to thank the reviewer again for their thoughtful feedback and suggestions. We have now performed additional analyses to address the issues raised, and have revised the manuscript accordingly (with revisions tracked in blue). We include a point-by-point description of the major changes below:
>
> ## Fine-grained analysis of failure modes
>
> We appreciate the reviewer’s suggestion to provide more fine-grained analysis of failure modes. We also agree that it is important to complement human behavioral evaluations with analysis by judges using a clear rubric. To address this, we have performed a fine-grained analysis of the images that received a low average agreement score in our human behavioral study (all images with an average agreement below 0.2, resulting in a total of 106 images). This analysis was performed by one of the authors, who classified these images according to four distinct failure modes:
>
> - Wrong subject: e.g., in the sentence ‘a rabbit chasing a tiger’, this error would indicate that the image does not contain a rabbit
> - Wrong object: e.g., in the sentence ‘a rabbit chasing a tiger’, this error would indicate that the image does not contain a tiger
> - Wrong relation: e.g., in the sentence ‘a rabbit chasing a tiger’, this error would indicate that the image does not contain a ‘chasing’ relation
> - Relation binding error: e.g., in the sentence ‘a rabbit chasing a tiger’, this error would indicate that the image depicts the roles reversed, i.e. the image depicts a tiger chasing a rabbit
>
> We found that the overwhelming majority of errors involved a wrong relation or a relation binding error, with a very small number of images that had the wrong subject or object. This result further underscores the difficulty that text-to-image models face with generating images based on relational prompts, and complements the findings from the summary agreement score provided by human participants.
>
> We have now included this analysis in the Appendix, section A.5.
>
> ## Assessment of inter-subject reliability
>
> We also appreciate the reviewer’s concern that a single summary rating might be an unreliable metric for evaluating text-to-image models. To address this, we calculated the inter-subject correlation for this metric. Specifically, we performed a test in which the participants were randomly split into two groups, and a correlation analysis was performed for the average scores for each image across the two groups. This test was repeated 100 times (with 100 distinct random splits) to estimate the average inter-subject correlation. This indicated a very high degree of inter-subject correlation (r = 0.91), suggesting that this a reliable metric for evaluating these models. It is also worth noting that, although participants provide only a single summary score for each image, this evaluation is not especially ambiguous or subjective (e.g. the image either includes the correct objects and relations or it doesn’t), which likely explains the high degree of inter-subject correlations for these ratings.
>
> We have now included this analysis in the Appendix, section A.6.

---

> ### Author Response · Authors · 2025-01-17
>
> ### Other issues:
>
> - We agree that it would be interesting to test more abstract or metaphorical statements. This would pose significant challenges for evaluation, i.e. how to determine what constitutes a ‘correct’ answer for a prompt like ‘happiness climbing a staircase of dreams’? In this work, our goal was to evaluate the more basic task of generating images based on concrete relational descriptions. Given that DALL-E 3 still faces some significant limitations in this domain, it is likely that it would struggle even more with more abstract prompts, but we agree that this is a very interesting question for future work.
> - We also agree that it would be interesting to look at prompts involving negation. A very recently published concurrent work [1] (which we have now added to the related work) suggests that this indeed poses a major challenge for these models. In this work, our focus was on drawing parallels between both text-to-image and image-to-text models, especially as this relates to the issue of compositionality, but we agree that logical constructs pose another interesting challenge that deserves further investigation.
> - It would be worthwhile to develop an automated approach for testing text-to-image models. However, our goal in this work was not to develop a generalizable method for automated evaluation, but instead to perform an evaluation (via human behavioral studies) of the current abilities displayed by text-to-image models. Given that the same difficulties are displayed by multimodal language models (as illustrated by our results in section 3, and the comparison of human-to-human vs. model-to-human correlations in the Appendix, section A.6), we believe that the current best approach for reliably evaluating text-to-image models is to use human raters.
> - We also agree that it would be interesting to extend the present study to investigate more specialized fields such as medical imaging. Based on our results, we anticipate that multimodal language models will have major limitations in these fields, as they seem to struggle with any task involving multiple objects and relations.
> - To address DALL-E 3’s more restrictive content filters, we revised any prompts that triggered content violations (described in section 2.1.1). The revised prompts did not trigger content violations, so this cannot explain the observed pattern of results.
>
> [1] Conwell, C., Tawiah-Quashie, R., & Ullman, T. (2024). Relations, Negations, and Numbers: Looking for Logic in Generative Text-to-Image Models. arXiv preprint arXiv:2411.17066.
>
> We would like to thank the reviewer once again for the very helpful comments and suggestions. We believe that the paper is stronger thanks to the inclusion of these additional tests, and look forward to any additional feedback or questions.

---

### Review · Reviewer_aDLR · 2025-01-04

**Summary Of Contributions:**

This paper investigated how the multimodal generative and understanding model performs for the compositional instructions. For generative models, the authors investigated how DALL-E 2/3 performs for instructions combining the objects with 1 or more relationships. They found that the model can perform worse when the objects are revised across the relationship connecting them, or they meet more complex compositional instructions. Similarly, they investigated whether the multimodal understanding model can correctly do classification for images given a compositional text prompt. In conclusion, they found that while these current models do have some capacity in compositional problems, they still fall short of human performance.

**Audience:**

Yes

**Claims And Evidence:**

Yes

**Requested Changes:**

1. Please add at least 2 more open-source models. For generative models, there are Mochi, CogVideoX. For understanding models, there are Qwen2-VL and Aria. And if with resources, Gemini and Claude can also be considered.
2. Refer to the second weakness.
3. Please improve the writing according to the thrid weakness.
4. Please explain the Spearman correlation according to the fourth weakness

**Strengths And Weaknesses:**

## Strengths
1. The paper provides a set of well-designed experiments to evaluate modern multimodal models (DALL-E 3, GPT-4V, GPT-4o) in compositional vision scenarios.
2. The authors have discussed the experiment results and concluded that while modern models do have some capability on compositional stuff, they still fall short of human performance.
3. The authors talk about some potential directions that can improve the model's ability on the compositional ability, which can serve as a reference for people in the community who are interested.

## Weaknesses
1. The evaluation only involves 3 models  (DALL-E 3, GPT-4V, GPT-4o) from the same company (OpenAI). Not a single open-source multimodal model have been evaluated on these experiments. This can cause some the potential bias in the results.
2. Most of the datasets used in the paper are simply re-purposed ones from previous works with little changes. There can be a lack of novelty since the results in this paper can already be reported by the authors of the original datasets. [1][2]  The authors need to clarify their differences with previous works the paper and highlight the unique contributions.
3. The writing of the paper needs to be improved. On page 10, line 5, (We used the original codebase from https://fleuret.org/ ...) the website URLs are explicitly written in the main paragraph instead of appearing as a footnote, which can affect the reading experience. Also it's better to have subsections in the related work section to help readers better understand the context in different topics.
4. The experiment number can sometimes not match the claim the authors have made. In second paragraph of section 2.2.2 Results, "We found that LL was significantly correlated with the agreement (spearman correlation r(598) = 0.090, p = 0.027)", the authors claim 0.09 as a high Spearman correlation, which is not the case. The authors did not explain where the magic number "598" comes from.


**References:**

[1] Conwell, Colin and Tomer David Ullman. “Testing Relational Understanding in Text-Guided Image Generation.” ArXiv abs/2208.00005 (2022): n. pag.
[2] Jiang, Huaizu, Xiaojian Ma, Weili Nie, Zhiding Yu, Yuke Zhu and Anima Anandkumar. “Bongard-HOI: Benchmarking Few-Shot Visual Reasoning for Human-Object Interactions.” 2022 IEEE/CVF Conference on Computer Vision and Pattern Recognition (CVPR) (2022): 19034-19043.

---

> ### Author Response · Authors · 2025-01-17
>
> We would like to thank the reviewer again for their thoughtful feedback and suggestions. We have now performed additional experiments to address the issues raised, and have revised the manuscript accordingly (with revisions tracked in blue). We include a point-by-point description of the major changes below:
>
> ## Evaluation of proprietary models from other labs
>
> We completely agree that it is important for the evaluation to include a more diverse set of models. We have addressed this in two ways. First, we have run Claude Sonnet 3.5 on the SVRT dataset (for which we could be most certain that neither the dataset nor similar variants were present in the training data). The results were qualitatively similar to those obtained with GPT-4v and GPT-4o. Specifically, we found that Claude Sonnet was affected by similar factors as human participants – i.e. performance improved as a function of the number of examples provided, and performance decreased as the number of objects in a rule increased – but the overall performance was nevertheless significantly below human participants. We did find that Claude Sonnet performed somewhat better than GPT-4o (with an overall accuracy increase of 4-5%), and performed better than chance on problems with 6 objects, but the overall pattern was very similar to that observed for GPT-4v and GPT-4o
>
> We have included these results in Figure 11 and section 3.2.2 of the revised manuscript.
>
> ## Evaluation of open-sourced models
>
> Second, we have now evaluated the open-source models QWEN2-VL-72B and InternVL2.5-38B on the SVRT dataset. We found that both of these models showed similar performance profiles to the GPT-4 models – i.e. they were both similarly sensitive to the number of examples and the number of objects – and both of these models performed significantly worse than human participants. The two open-source models also performed at a similar level as GPT-4o (i.e., there were no statistically significant differences between these models).
>
> These results are also now included in Figure 11 and section 3.2.2.
>
> ## Clarification of novelty and major contributions
>
> We have now added a section to the paper clarifying the novelty and unique contributions of our work. In particular, we have generated major extensions of the dataset from [1] for evaluating text-to-image models, including the generation of the datasets for the reversed and compositional prompts, which were crucially important in exposing the persistent weaknesses of current models. We also performed extensive human behavioral evaluations to assess the performance of the text-to-image models, and to establish a baseline with which to compare the multimodal language models (e.g., in the experiments with SVRT). Finally, we have synthesized the results of these diverse experiments to identify a common and persistent weakness displayed by all current multimodal generative models (including both text-to-image and image-to-text models) with understanding and generating compositional scenes.
>
> We have included an explicit summary of these contributions at the end of section 1.
>
> [1] Conwell, C., & Ullman, T. (2022). Testing relational understanding in text-guided image generation. arXiv preprint arXiv:2208.00005.

---

> ### Author Response · Authors · 2025-01-17
>
> Other issues:
>
> - We have now moved the website URL to a footnote as suggested.
> - We have now divided the Related Works section into subsections.
> - We have revised the description of the correlation analysis for reversed prompts to clarify that it is statistically significant, but with a modest effect size. The analysis has df=598 because the correlation is performed over scores for N=600 images, and degrees of freedom for a correlation analysis is equal to N-2.
>
> We would like to thank the reviewer once again for the very helpful comments and suggestions. We believe that the paper is stronger thanks to the inclusion of these additional tests, and look forward to any additional feedback or questions.

---

### Decision · Action_Editor_Yq7N · 2025-03-13

**Recommendation:** Accept as is

**Comment:**

This paper examines the performance of multimodal generative and understanding models when handling under-compositional instructions. Reviewers agree that the study explores an important topic and features well-designed experiments. The authors have addressed concerns by incorporating additional results from various model families (e.g., Claude), open-source models (e.g., QWEN2-VL-72B), and providing necessary clarifications.

I encourage the authors to integrate these results and clarifications into the final version of the paper.

**Audience:**

This paper explores multimodal generative models in the context of complex scene understanding. Therefore, I believe it will be of interest to both the natural language processing and computer vision communities.

**Claims And Evidence:**

All claims in this paper are thoroughly backed by empirical results and supporting evidence.